# Contribution to the Study of Forest Fires in Semi-Arid Regions with the Use of Canadian Fire Weather Index Application in Greece

**Nikolaos Ntinopoulos, Marios Spiliotopoulos \***  **, Lampros Vasiliades**  **and Nikitas Mylopoulos**

Laboratory of Hydrology and Aquatic Systems Analysis, Department of Civil Engineering, University of Thessaly, 383 34 Volos, Greece
*   Correspondence: spilioto@uth.gr

**Abstract:** Forest fires are of critical importance in the Mediterranean region. Fire weather indices are meteorological indices that produce information about the impact as well as the characteristics of a fire event in an ecosystem and have been developed for that reason. This study explores the spatiotemporal patterns of the FWI system within a study area defined by the boundaries of the Greek state. The FWI has been calculated and studied for current and future periods using data from the CFSR reanalysis model from the National Centers for Environmental Protection (NCEP) as well as data from NASA satellite programs and the European Commission for Medium-Range Weather Forecasts (ECWMF) in the form of netCDF files. The calculation and processing of the results were conducted in the Python programming language, and additional drought- and fire-related indices were calculated, such as the standardized precipitation index (SPI), number of consecutive 50-day dry periods (Dry50), the Fosberg fire weather index (FFWI), the days where the FWI exceeds values of 40 and 50 days (FWI > 40) and (days FWI > 50). Similar patterns can easily be noted for all indices that seem to have their higher values concentrated in the southeast of the country owing to the higher temperatures and more frequent drought events that affect the indices' behavior in both the current and future periods.

**Keywords:** forest fire; Greece; Canadian fire weather index; Fosberg fire weather index; Greece; Mediterranean; agrometeorology; droughts



## 1. Introduction

The Mediterranean climate tends to create ecosystems that exhibit increased aridity, high temperatures, and extreme events during the summer season. It is estimated that on an annual basis, roughly 50,000 forest fire incidents occur in the region, leading to the burning of an estimated 470,000 ha area [1]. Most forest fires in the Mediterranean region can be traced to human activity, which is supposed to be the leading factor for the appearance of forest fires in the broader region. Forest fires pose a complex threat to modern societies and are a threat comprised by environmental, socioeconomic, and, above all, human costs. For all these reasons, it is better to avoid forest fires that result in them. It is hence of critical importance that fire agencies possess adequate means for the prediction and thus the prevention of low-, medium-, and high-scale fire events. Fire danger indices as well as the application of fuel treatment projects are examples of such means towards fire prevention and suspension assistance, although typical Greek National policy is heavily suspension oriented, with more funds and effort traditionally being allocated towards post-ignition fire suspension, an approach that can fail when multiple fire events occur [2].

The occurrence of forest fires, although present in the broad Mediterranean region for quite a significant time span, is expected to increase in terms of both the general number of occurrences as well as in their overall consequences in ecosystems due to climate change [1]. The results of Weber et al. [3] found current trends in the Mediterranean climate, more

specifically in Greece, to indicate ever increasing intensities of drought events that even extend out of season. The Mediterranean Basin and particularly Greece present an increased vulnerability regarding climate change impacts [4]. As a result, deriving the link between fire weather and fire occurrence within the region is a continuously increasing challenge.

A projected increase in temperature is expected to lead to a decrease in the relative humidity, thus incrementing soil fuel dryness. According to Merril et al. [5], this effect will worsen in regions with lower rainfall. Forest fires with behavior closely tied to fuel moisture present increased sensitivity regarding climate change effects [3]. Greece therefore presents increased vulnerability regarding fire increase and climate change in general [4] since it is located in the eastern edge of the Mediterranean Basin.

Historically, forecasting of fire danger was mainly conducted via statistical techniques exploiting the lagged relationships between different fire statistics (number of fires, total burned area) [6]. Additionally, forecasting was based on meteorological observations at global to regional scales. Empirical approaches, however, insert limitations caused by the short history of observation-derived data, with inaccuracies and the point observation data not being representative of the stations in broader region stations considering losing as the result of local small-scale differentiations in meteorological variables. Modern advances in climate science and climate models of atmospheric and oceanic parameters have made the generation of numerical climate models (Global Climate Models, GCMs) possible, producing dynamical simulation-based predictions on different time scales [7] and offering an alternative to empirical approaches [8]. Forecasts that are the outputs of model simulations are unavoidably accompanied by a degree of uncertainty since the outputs coming from GCM results present significant deviations from the observed climate due to model biases [9,10]. To outweigh the effect of such errors accompanying model simulations, the mean outputs of multiple models are often selected and are called ensembles [8]. Being composed of many different models runs of such ensembles is computationally intensive, and hence, these models are run at lower resolutions than a single deterministic run. The forecast is then interpreted as probabilistic rather than deterministic. Ensembles can boost confidence in the decision process during emergency situations, as a cost–loss analysis can be associated with the different scenarios [11].

Out of the many fire indices being in use in the literature [12], the most frequently used one is the Canadian fire weather index (FWI) [5] developed by the Canadian Forest Fire Danger Rating System (CFFDRS). The FWI is applied in this study for the region of Greece to estimate fire danger in both current and future time periods and to find a link between FWI and the climate regime of the region. According to Schlobohm et al. [13], the fire weather index is more properly classified as a fire danger rating system than as a fire index. Fire danger rating systems incorporate methods and models for data gathering and processing to convert inputs into numerical qualitative ratings. These danger ratings are often ranked from low to extreme and are visually plotted on maps that are user friendly and easy to interpret [14]. Systems such as the FWI detect and assess weather favorability over fires and their potential intensity as well as spread rather than provide an estimate on the probability of occurrence itself and general fire behavior. The FWI (developed in Canada) is specifically calibrated to describe the fire behavior in jack pine stands (*Pinus banksiana*), which are typical of Canadian forests. The FWI can be implemented in a wide array of ecosystems [15,16]. In addition, Ziel et al. [17] compared the Finnish forest fire index (FFI) to the FWI in Finland. They concluded that both indices performed similarly over southern and central Finland, but the FWI identified more high-danger incidents than the FFI in northern Finland [18]. Nunes et al. [19] employed two models using the influence of meteorologic-climatic factors through the DSR to predict fire danger in Portugal based on the exceedance probability of the burnt area thresholds in the region. Calheiros et al. [20] also found the DSR to be an effective indicator of burnt area variations. Mestre et al. [21] found that the implementation of the FWI can provide an adequate medium for the development of an early warning system regarding wildfires.

FWI calculation relies solely on meteorological variables and no information on the actual vegetation state, ridding its calculation of a degree of complexity [15]. However, as previously stated, the FWI is not a physical measure of fire activity, but of its potential danger given an ignition event. Therefore, high values might not be the lone result during ignition in the absence of an ignition source or event, nor can they function as a guarantee of significant wildfire effects if suppression is effective and properly timed [22]. This should be taken into account by the user (usually fire agencies) when studying wildfire behavior and effects, for which case inclusion of indices that take into account landcover, land use, and vegetation of the ground is considered better practice. Table 1 shows a classification scheme of the values of the FWI index. The current paper examines the spatial patterns of the FWI as well as the link with topography through meteorological influence for the Greek spatial domain.

**Table 1.** FWI value categorization.

| FWI | Danger Level |
| --- | --- |
| 0–5.2 | Very Low |
| 5.2–11.2 | Low |
| 11.2–21.3 | Moderate |
| 21.3–38.0 | High |
| 38.0–50.0 | Very High |
| ≥50.0 | Extreme |

Previous studies have shown that in Mediterranean-type ecosystems, climate plays a key role in influencing the interannual variability of fires by affecting the fuel-layer structure and moisture [23–26]. Weather and climate are the most prominent predictors when it comes to fire occurrence [27], with weather affecting fuel-moisture conditions during the fire season and climate influencing the fuel availability and composition of an area [28]. The results from the study [29] show that the FWI values do not present a significant link with Burned Area and the number of fires; however, the FWI percentiles present an exponential correlation with the latter, with 40% of fires occurring for FWI90 (90th percentile FWI), 65% occurring for FWI 75, and a total 96 of the fires being observed for days with FWI25. Dimitrakopoulos et al. [30] proposed a new classification scheme regarding FWI values for Mediterranean ecosystems due to the existing Canadian thresholds not being able to accurately reflect trends in fire occurrence. According to Bedia et al. [31]. FWI was a good forecast method for high fire danger for the fire events of 2003 and 2007 in Greece. Pinto et al. [32] modified the FWI to also incorporate the CHI (continuous Haines index), to create an index called effective FWI (FWIe), which better accounts for atmospheric instability and is better able to predict the energy levels released by fire. The index was studied and showed accurate results in the case studies of Moncique and Guadalajara. Sirca et al. [33] found the FWI and IFI (integrated fire index) to be the best-performing ones regarding fire predicting capability in the study regions of France and Italy. Urbieta et al. [34] also found the FWI to be highly effective in fire prediction in five southern EU countries and on the Pacific coast of the USA. Junior et al. [35] performed automatic calibration of the index for 769 regions within Europe to more accurately define fire danger and predict fire behavior.

In order to calculate the FWI, temporal and spatial patterns and correlations between indices were extracted, and the python programming language was used. Data for the FWI came in the form of .csv files containing climate variables, which are outputs of the CFSR model in SWAT file format and as netCDF files from NASA and the ECWMF for the current and future periods, respectively. Extracting correlations between indices and the factors related to climate such as subindices of the FWI system and topography can to ensure better comprehension of the effect that these two factors have on the appearance and vulnerability regarding wildfires within the boundaries of the Greek domain.

## 2. Materials and Methods

### 2.1. Study Area

Greece is located at the most southeastern edge of Europe and is located at latitudes from 35°00′ N to 42°00′ N and at longitudes from 28°00′ E to 30°00′ E [12]. The majority of Greece's geographic boundaries comprise coastline (Figure 1). The Greek climate is classified as the Mediterranean climatic type and is characterized by mild winters, relatively warm and dry summers, and a long sunshine duration for most of the annual cycle. The climate of Greece varies from continental Mediterranean in the north to subtropical Mediterranean in the south, with various intermediate climate subzones [12]. The increased variability in the Greek climate can be attributed to the effects of topography on air masses coming from the central Mediterranean. This results in the western part of the country being more humid compared to the drier and hotter eastern part [36] being much drier and windier during the summer season. The Greek fire season covers the month range of May–October, with the rest of the months mainly being characterized by mild winters that contribute to the production of biomass, which acts as a fuel layer after losing moisture during the summer period. The main forest types are composed of Pinus Halapensis and Pinus Brutia [22]. The mean elevation of the country is 498 m, and Greece presents quite rough topographical features. Studies have shown that in Mediterranean ecosystems, climate is the main seasonality factor regarding wildfire occurrence due to it having a strong effect on the composition and flammability of the fuel layers on the ground [37], with increases in the latter being considered the main response of ecosystems to climate change [38].

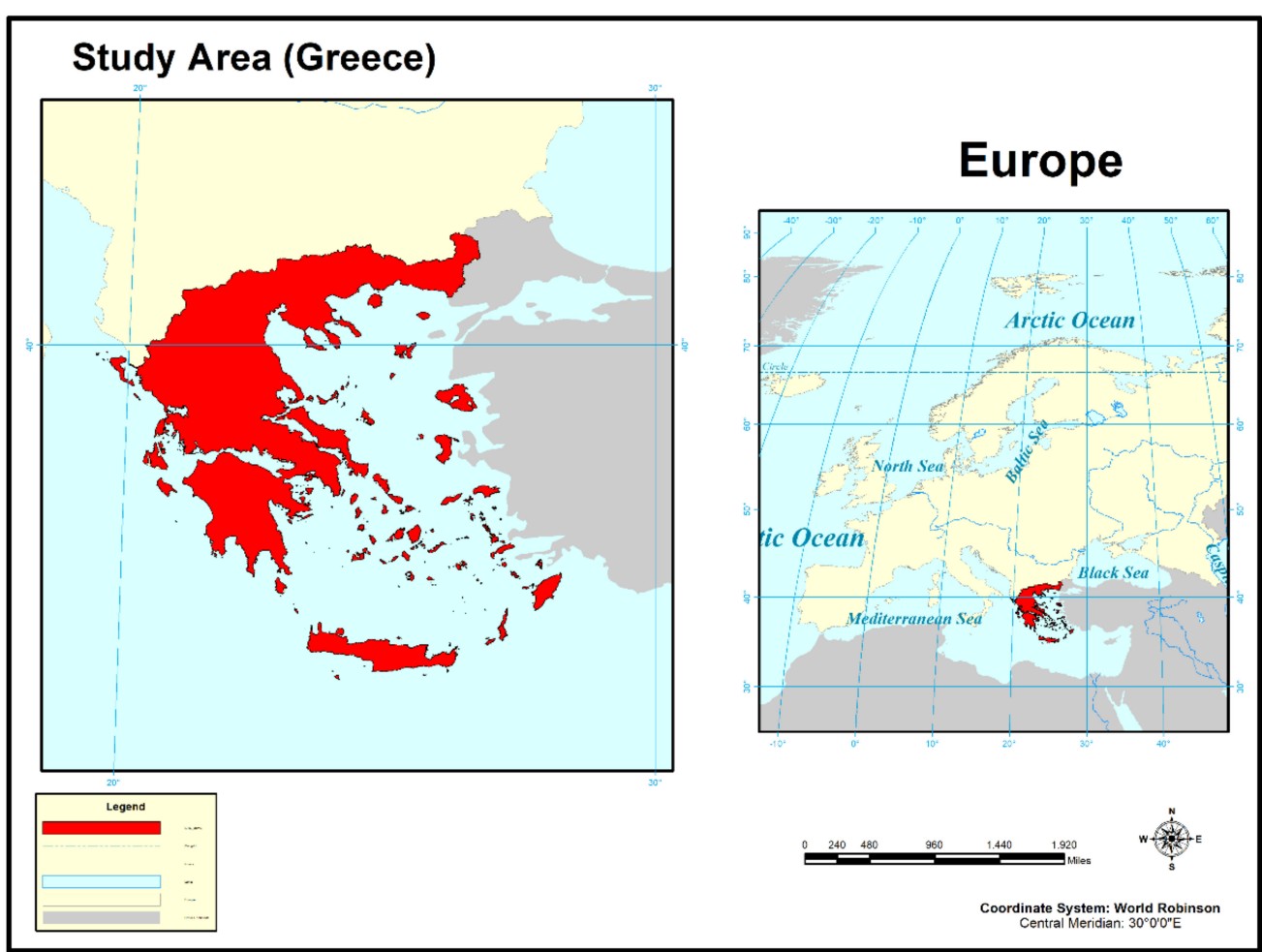

**Figure 1.** Depiction of the study area.

*2.2. Data Acquisition*

Data for the calculation of the FWI were obtained from (https://swat.tamu.edu/data/cfsr (accessed on 18 March 2022)) in the form of .csv SWAT input files. The data were produced by the CFSR (Climate Forecast System Reanalysis), which is a global, high-resolution, coupled atmosphere–ocean–land surface–sea ice system. The CFSR reanalysis model implements the coupling of oceanic and atmospheric parameters to achieve the generation of 6 h guess fields. Systems employed by the CFSR include:

- Operational Global Data Assimilation System (GDAS);
- Atmospheric GADAS-Gridded Statistical Interpolation (GSI);
- Ocean–ice GODAS;
- Land GLDAS;
- Atmospheric Model: operational Global Forecast system (GFS);
- Ocean Model: MOM4 Ocean (GFDL Modular Ocean Model);
- Land Model: operational Noah Land Model;
- Sea Ice model from the GFDL Sea Ice Simulator.

The .csv files of the dataset bear daily information about wind speed (m/s); precipitation (mm); relative humidity (fractional); minimum, mean, and maximum temperature (°C); and solar irradiance (W/m$^2$). Each .csv file represents a modeled point at a spatial resolution of ~38 km (T382). The total spatial coverage of the dataset is 0° E–359.687° E longitude and −89.761° N–89.761° N latitude. To isolate the study area, a bounding rectangle was selected to cover the Greek domain, with a total of 339 .csv file points.

Data from NASA's Goddard Space Flight Center (https://portal.nccs.nasa.gov/datashare/GlobalFWI/v2.0/ (accessed on 18 March 2022)) were also used in the form of netCDF files. The mentioned files contained monthly values for the component indices of the Canadian FWI system calculated by NASA's Modern-Era Retrospective analysis version 2. Four datasets from NASA satellite missions and algorithms were used: IMMERG.v6 long-term mean, GPM.v5-FINAL, TRMM.v5 long-term mean, and GPCP long-term mean, and cover the periods 2001–2019, 2018, 1998–2014, and 1997–2014, respectively. For each one of the datasets, netCDF files with monthly values were acquired and were then merged and averaged for the time period covering period of March–October.

For the study of future predictions made by the index, the data came in the form of netCDF files provided by the *European Commission for Medium Weather Forecasts (ECMWF)* (https://cds.climate.copernicus.eu/cdsapp#!/software/app-tourism-firedanger-indicators-projections?tab=app (accessed on 18 March 2022)) for the full time periods of 2041–2060 and 2079–2098. For future projections of the index's RCP (Representative Concentration Pathway) scenarios, data with values of 4.5 and 8.5 were selected to represent the mean and worst case, respectively. The data came as the mean output of an ensemble model comprising the following climate simulation models: *CNRM-CM5 (CNRM-CERFACS, France), EC-EARTH (ICHEC, Ireland), IPSL-CM5A-MR (IPSL, France), HadGEM2-ES (UK Met Office, UK), MPI-ESM-LR (MPI, Germany), and NorESM1 m (NCC, Norway)*. This database was excluded when studying the current period's FWI due to its historical dataset only reaching as far back as 2005.

Data for 8 meteorological stations were also used to act as validation for calculations alongside the NASA-derived datasets. These stations were the ones for the cities (Athens, Drama, Nevrokopi, Rethymno, Alexandroupolis, Florina, Volos) in the "CLIMPACT" network (https://climpact.gr/main/network (accessed on 18 March 2022)), which is a joint multilateral incentive between eleven scientific institutions in Greece that aims for cooperation with the National Committee for Climate Change to provide insight and consultation regarding climate-related matters. Data for said stations covered the period of 2010–2020 in the form of .xls files, with days/rows with empty observations being discarded from computations.

*2.3. FWI Calculation and Structure*

The FWI index is calculated on a daily basis from the values corresponding to the previous day. The FWI is a pretty complex procedure that requires the implementation of equations derived by van Wagner et al. in order for it to be calculated [37]. The calculation of the values of the Canadian FWI system was performed in the Python 3.8 programming language according to the code provided by Wang et al. [38]. Further study and modification of the results for both present and future periods were also performed in Python 3.8. The FWI system comprises 5 subindices (Figure 2). Three of those indices are moisture codes numerically describing the moisture content of a specific fuel layer of the ground, whereas the latter two are behavioral indices assessing fire growth and spread.

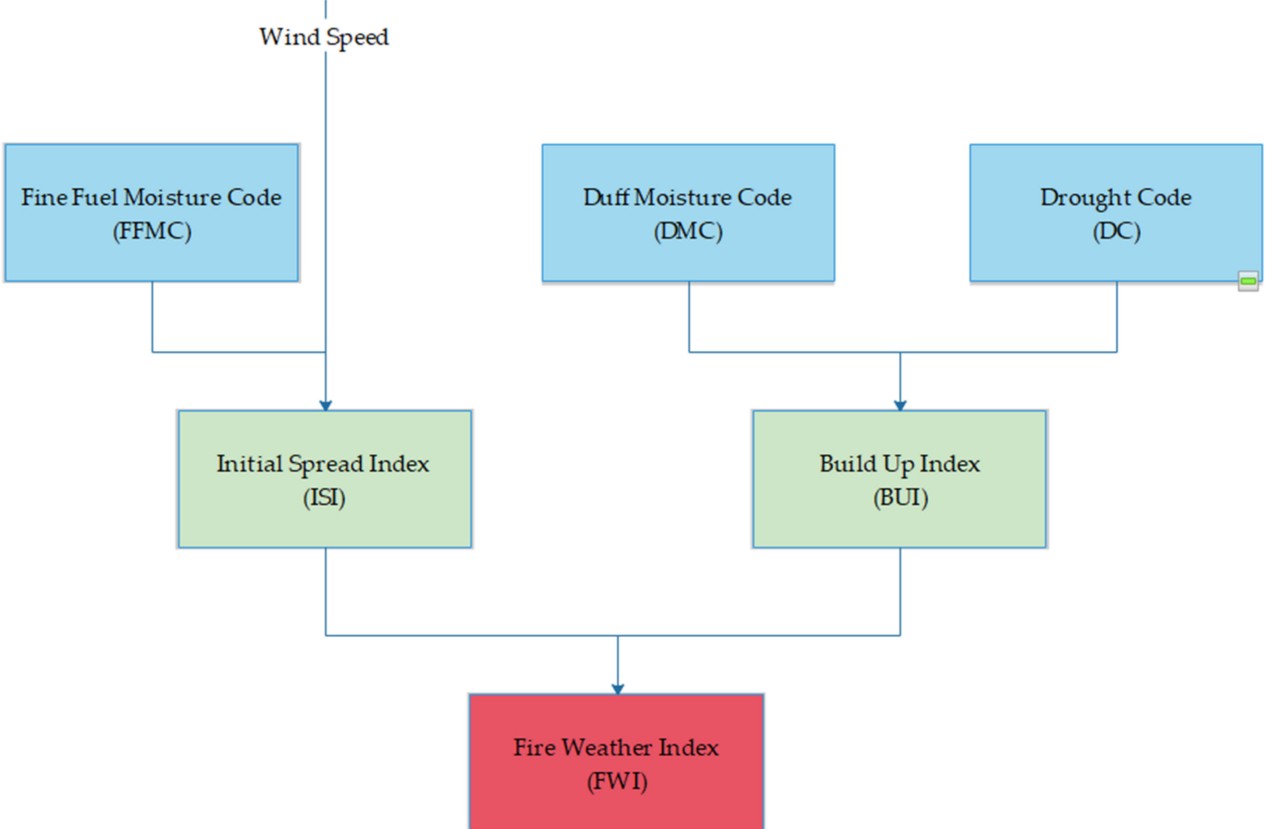

**Figure 2.** Schematic representation of FWI scheme.

**The Fine Fuel Moisture Code** (FFMC) refers to the moisture content of litter and other cured fine fuels occupying the upper fuel bed layers (surface layers, 1–2 cm deep). This code indicates the relative ease of ignition and the flammability of fine fuel, and it is characterized by a fast response to weather variations, with a time-lag of 16 h under standard conditions. FFMC is a unitless number ranging between 2 and 101, with an initial conventional value of 85.

**The Duff Moisture Code** (DMC) is a moisture code describing the moisture content of the fuel layers found at the depths of 5–10 cm (duff layer). DMC is characterized by a medium-term response (10–12 days) to weather variations. Its value is also unit-less, and it ranges in the spectrum from (0 to +∞, with its conventional initial value being 6.

**The Drought Code** (DC) is the last moisture code of the system and describes the moisture content of the deeper and most compressed fuel layers (10–20 cm). This code accounts for the effects of seasonal drought as well as for the effects of smoldering in the deep duff layers and in logs. The code presents a long-term response (50 days) regarding weather variations, ranging between 0 and +∞ and having a conventional initial value of 15.

**The Initial Spread Index** (ISI): This index is a unitless numeric measure used to assess fire spread rate at early stages accounting only for FFMC and wind speed. It ranges in the spectrum (0, +∞).

**The Buildup Index** (BUI): A numeric unitless index rating the availability of fuel for potential combustion. This index is a weighted combination of DC and DMC and also ranges from 0 to +∞.

**The Fire Weather Index** (FWI): The final result of the FWI system is an index with no units that rates the general intensity of a fire. It is the end result of the integration of the prior mentioned subindices and belongs to the range (0, +∞).

**The Daily Severity Index** (DSR): A nonlinear transformation of the FWI that can provide additional information regarding the difficulty of suppression of a wildfire. It can be averaged over time giving the Seasonal Severity Index (SSR) and space.

Two more indices were calculated together with the existing codes of the FWI system. These were the days with FWI > 40 and days FWI > 50, which were calculated in order to track the number of days within the fire seasons (March–October) of the five-year period (2010–2014) covered by the CFSR dataset. The cases where the FWI surpassed these critical values can be classified as very high and extreme, respectively, according to the ECMWF.

The CFSR dataset provides the option to see the min, max, and mean temperature of the simulated day (Table 2). The FWI system is calibrated and optimized for meteorological values representing the conditions at 12 pm noon [39]. Given the absence of such measurement, the max temperature of the dataset was selected to account for the loss of extreme events, meaning days with FWI > 50 due to the acquisition of the mean value of the index for each point for the five-year period. Hence, the average values of the FWI might, at first glance, seem small compared to the ones expected for a Mediterranean ecosystem; however, they represent the mean situation for each pixel and correspond, as will be later shown, to comparatively large values regarding the number of days above the mentioned thresholds of FWI 40,50. After calculating the daily values of the FWI system for each point and obtaining the mean values for each code as well as the added indices, the output data were converted to a Python GeoDataFrame that was then converted to a point shapefile containing the mean values for the indices mentioned for each of the 339 points of the dataset. The Kriging method was then applied in ARCGIS 10 to obtain thematic maps for each index using a shapefile with Greece's geographic boundaries obtained from (https://www.diva-gis.org/ (accessed on 18 March 2022)).

**Table 2.** Meteorological data used for the composition of the FWI from the CFSR dataset covering the period of May–October in 2010–2014.

|         | Temperature | Relative Humidity | Wind Speed | Precipitation |
|---------|-------------|-------------------|------------|---------------|
| count   | 339         | 339               | 339        | 339           |
| mean    | 14.021811   | 0.391961          | 2.091348   | 1.057967      |
| std     | 1.126246    | 0.025283          | 0.913227   | 0.754108      |
| min     | 10.227636   | 0.312598          | 0.96965    | 0.110498      |
| max     | 17.757304   | 0.459695          | 4.23243    | 4.073488      |

*2.4. Future Projections*

To study of future projections made by the FWI, the data were obtained by the ECMWF in the form of netCDF files. The temporal coverage of the files are the full periods of 2041–2060 and 2079–2098 for the RCPs of 4.5 and 8.5. The datasets encompass a broad region, including the European domain; thus, the boundaries had to be specified via coordinates that capture the geographic domain, including Greece, more specifically. The dataset's coordinates come in rotated North Pole coordinates, and in order for the region of Greece to be portrayed in more detail, the coordinates needed to be specified as rlat = 39.5, rlon = 128. The dataset includes but is not limited to information for the number of days per pixel for each simulation period, with the FWI projected to surpass the three thresholds that have been set by the ECMWF, with 15, 30, and 45 defining three distinct fire danger

classes: low, medium, and high fire danger, respectively, for both RCPs. The Mann–Kendall test was performed for a pair of points: rlat = −10, −5, for the rotated pole coordinates of rlon = 1, 2, 3, and 4, for the two time periods for the RCPs of 4.5 and 8.5. The aim was to obtain trends for the series of the FWI in those two future modeled time periods.

### 2.5. Additional Indices

Additional indices were also calculated to study the correlation between the FWI and other fire indices, such as the Fosberg fire weather index (FFWI). Drought was studied through the Standardized Precipitation Index (SPI) as well as by the number of 50 consecutive day periods without rain (codenamed Dry50). Landsat 7 NDVI was also used to correlate FWI values with canopy health and hence with fuel moisture.

The FFWI is a meteorological fire weather index that only relies on information about the temperature, relative humidity, and wind speed. Similar to the Canadian FWI, it too does not require fuel-related information and offers the advantages of having very simple calculation equations. The results of the index have been shown to be accurate in classifying fire danger levels [40–42]. Values for the FFWI can range between 0 and 100, with increasing values revealing increasing favorability of meteorological conditions towards fire occurrence and spread. Index values ranging between 20 and 30 indicate a high danger for fire spread, with the ones exceeding 30 being characterized as representing very high to extreme levels of danger. The upper threshold of the index has the value of 100, corresponding to conditions with zero relative humidity and a wind speed of 30 mph. Values exceeding the threshold of 100 are set to the said value. The FFWI is a non-linear combination of linearly related data obtained for temperature, relative humidity, and precipitation [43]. Assumptions by Fosberg et al. [44] made for fuel properties regarding index calculations consider the ratio of the fuel bed surface area to the volume as a constant value in space and time. The index calculation formula is, in essence, divided into two fundamental components: one concerning fuel moisture and one related to rate of spread [44], with the two components being based on the models of Rothermel et al. [45] and Simard et al. [46] accordingly.

Equations (1)–(3) present the calculation scheme of the FFWI.

$$\text{FFWI} = \eta \sqrt{1 + U^2}/\text{O.3002} \tag{1}$$

where U is the wind speed in mph, and η is the moisture damping coefficient calculated according to Equation (2).

$$\eta = 1 - 2\left(\frac{m}{30}\right) + 1.5\left(\frac{m}{30}\right)^2 - 0.5 \, (m/30)^3 \tag{2}$$

Finally, the equilibrium moisture content m is a function of the temperature (T) in degrees Fahrenheit and the relative humidity (Rh) in percent.

$$m = \begin{cases} 0.3229 + 0.281073 \, h - 0.000578 \, Rh \, T \, , \ Rh < 10\%, \\ 2.22749 + 0.160107 \, Rh - 0.01478 \, T \, , \ 10\% \leq Rh \leq 50\%, \\ 21.0606 + 0.00556^2 - 0.00035 \, Rh \, T \ - 0.483199 \, Rh \, , \ Rh > 50\% \end{cases} \tag{3}$$

The fuel properties of the FFWI model do not describe any particular fuel type. Studied fuels are assumed to be very fine (surface area/volume = 3000 ft$^{-1}$), with a moisture extinction value of 30% [43].

The standardized precipitation index (SPI) developed by McKee et al. [47] is a simple meteorological index that relies on monthly cumulative precipitation. The index is calculated as the difference of the total precipitation for a given month subtracted from the mean value of precipitation for this particular month over the calculation period (1–48 months). Depending on the calculation timescale, the SPI can be related to the soil moisture of the smaller time scales and to groundwater and reservoir recharge for the longer ones. The results of the index are given as units of standard deviation from the long-term mean. Values of the index range between −2 and 2, with decreasing values signifying drought

conditions for the studied location, although higher ranges can also be observed. The historic precipitation record is fitted to a probability distribution that is then converted to a normal distribution, meaning a mean SPI of zero for any examined location. Since the SPI is measured in units of standard deviation from the mean, the index can be useful for portraying and comparing precipitation anomalies for any given location at different time scales. Indicators next to the index SPI-1–SPI-48 indicate the period in months for which the calculations take place [48].

- **SPI 1-3:** An indicator of the immediate impacts of precipitation shortages in soil moisture, snowpack, and runoff in small creeks.
- **SPI 3-12:** Indicator regarding stream flow and ground water reservoir storage.
- **SPI 12-48:** Indicator for reduced reservoir and groundwater recharge.

In the current study, the SPI-1 was calculated to assess the short-term impacts of precipitation shortages in soil moisture for the modeled points.

The number of 50-day dry periods was taken as the number times during the time period when a sequence of 50 or more days appeared with 0 mm of precipitation, with the count increasing each time the period is interrupted by a day with precipitation > 0 mm. All meteorological indices were calculated for the 2010–2014 period with data from the CFSR dataset, which were later correlated with the FWI.

By importing the end-point shapefile to Google Earth Engine (GEE), the NDVI was calculated and extracted for each one of the 339 points using reflectance values from the Landsat 7 "*USGS Landsat 7 Collection 1 Tier 1 and Real-Time data Raw Scenes*" dataset (https://developers.google.com/earth-engine/datasets/catalog/ANDSAT_LE07_C01_T1_RT (accessed on 18 March 2022)). The index was calculated for the period of 2010–2014 and had a spatial resolution of $30 \times 30$ m when using the near-infrared band (NIR) "B4" in the wavelength range of 0.77–0.90 μm and the red band "B3" in the wavelength range of 0.63–0.69 μm. The NDVI is calculated as the normalized difference between the NIR and Red bands, with values ranging from −1 to 1, with higher values approaching 1 indicating an increasing moisture content in plant tissues and hence more canopy coverage for the examined grid cell. A healthy canopy tends to have high values of reflectance on the NIR spectrum due to the presence of chlorophyll, with low health or dead plants reflecting more on the red spectrum and less on the NIR spectrum, shifting the index values towards 0. Values of the NDVI closer to −1 indicate the presence of water bodies. Studies have shown that NDVI bears high effectiveness in differentiating forest types and agricultural field properties [49] as well as other vegetation properties, including the Leaf Area Index [50], biomass [51], chlorophyll content [52], plant yield [53], the vegetation cover of a region [54], and plant stress–health [55]. Such estimations are often derived by correlating remotely sensed NDVI observations coupled with the ground-measured values of these variables.

Despite the NDVI image having a much higher resolution than the meteorological values obtained for the CFSR dataset, the outputs of the index were clipped to the shapefile for each point specifically. A condition was set in the GEE code for the parameter 'CLOUD_COVER' to be less than 5%, and mean values of the index were selected for the study period to study a possible relationship with the FWI values.

All of the procedures and methods for this study are depicted in the flowchart in Figure 3.

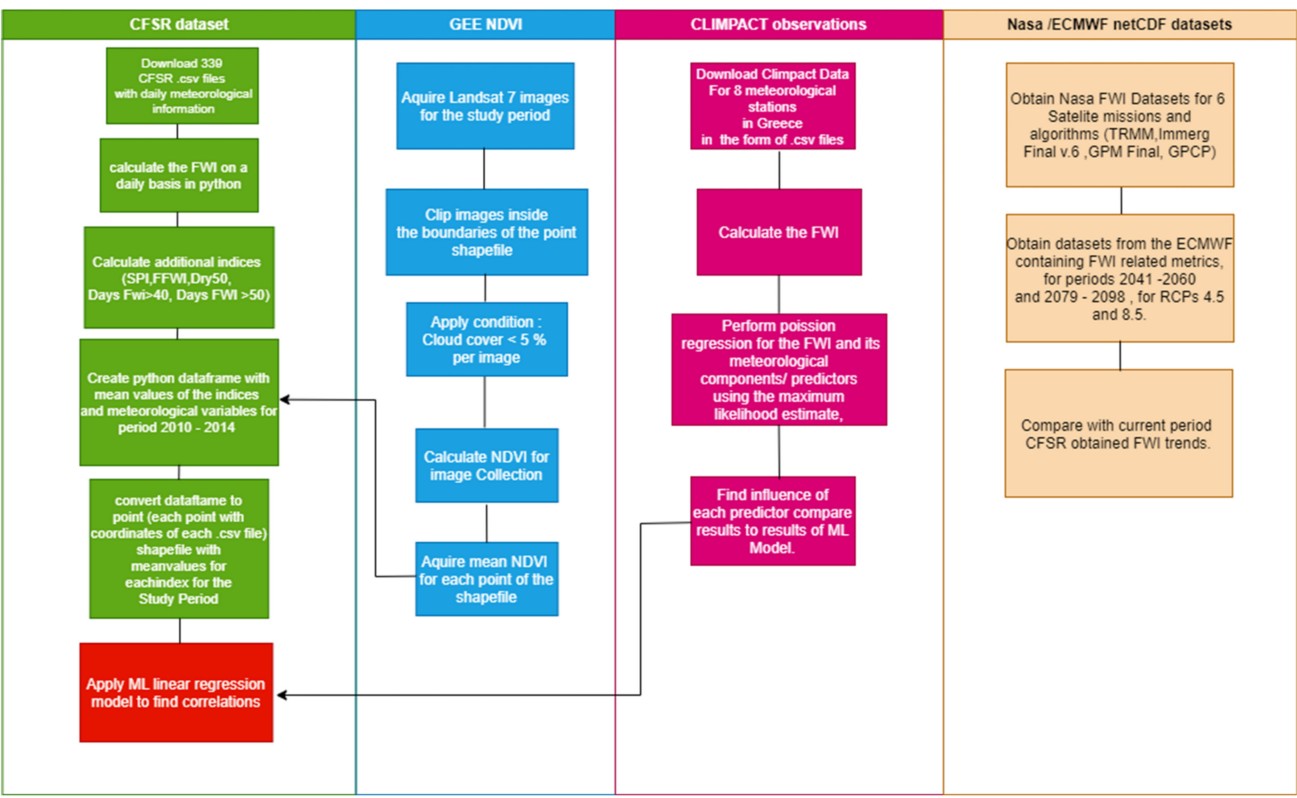

**Figure 3.** Detailed flowchart of procedures.

### 2.6. Use of Machine Learning Linear Regression Model to Find Correlations between Indices

Despite the non-linearity of the FWI calculations regarding its meteorological components, a linear regression model was opted to be implemented to study the relationship between the FWI, its meteorological components (temperature, wind speed, relative humidity, and precipitation), as well as the other mentioned indices, due to the similarity regarding spatial distributions. The model was written in Python programming language. A total of 100 model runs were performed, the mean results of which are displayed in this study. During each run of the model, the aim is to find a correlation between the FWI itself and its related indices. The model implements the linear regression method ten times for each run, each time for one of the related indices (FFWI, SPI, days FWI > 40, days FWI > 50, dry50, temperature, wind speed, relative humidity, precipitation, and NDVI).

Each time a run of the series bears one of the mentioned indices, it acts as the independent variable (input X), whereas the FWI series always plays the role of the dependent one (output Y). The model splits the data series into training and test sets. The training set is used for the training of the model in order to recognize patterns in the data and to obtain the best possible linear correlation (y = a * x + b), while the test set is used for predictions after obtaining the correlation. The data have been empirically split using the train/test analogy of 65%/35% for the data each time. Higher percentages of the sample dedicated to the training set will lead to better quality training of the model. However, it is advised that splits overly favoring the training sample be avoided to prevent the model from being "over-fitted" to the data, with the term referring to a state where the model is extensively and specifically trained to a particular dataset, thus being in danger of being rendered inaccurate and ineffective by a slightly similar or noise-including dataset. The model finally includes a score function that returns a numerical rate of the simulation, that being the $R^2$ value.

*2.7. Examination of the Influence of Meteorological Variables on FWI a Daily/Single Point Basis*

When examining the FWI against its predictors and meteorological variables, correlations appear weak with $R^2 < 0.5$ for all predictors on a time series scale. For this reason, the effect of each predictor of the FWI was examined based on the index's sensitivity to its alteration, namely each time component would be chosen to act as the influencing variable, with the other three being set as constants in their mean values for the time series, except precipitation, which was set to 0. The varying component predictor would range from 0 to its maximum value, with intervals being fourth fractions of its maximum value. The effect of each predictor portraying the sensitivity of the FWI on it was taken as the absolute value of the percentage difference of the FWI as shown in equation (4).

$$FWI\ sensitivity\ (x) = \frac{|FWI_{xi-1} - FWI_{xi}|}{FWI_{xi-1}} \qquad (4)$$

with $FWI_{xi}$, $FWI_{xi-1}$ being the values of the index for the $i$th and $i$th $-1$ values of the predictor variable being different fourth fractions of the maximum value for said predictor in the time series.

The effects of precipitation on the values of the FWI were also examined for both the mean and maximum fixed values of the other predictors, except relative humidity, which was kept as its mean value to test the effect of precipitation against maximum favoring conditions (high temperature and wind speed). The index was calculated and plotted for an array of artificial daily precipitation values that ranged from 0.1 to 40 mm, with intervals of 0.1 up until the value of 1 mm, of 1 until the value of 11 mm, and 10 from the value of 20 until the final value of 40 mm. Then, to most accurately track the influence of the predictor on the index values, the FWI the index was calculated and studied across the span of 35 days again under the same conditions and with precipitation being set to zero, except for the value of the 15th day. The experiment aimed to study the effect a single day of precipitation on the index value as well as the number of days the index takes to recover, with value recovery being defined as $FWI_{post\ precipitation} \geq 0.95 * FWI_{pre\ precipitation}$. The aforementioned experiment was performed for distinct precipitation events of (1, 5, 7, 10, 20, 40 mm) and was then repeated with each value being distributed across two, three, and then four consecutive days, with them being the 15th–16th, 15th–17th, and 15th–18th accordingly.

## 3. Results

*3.1. General Overview*

A common pattern seems to occur for all datasets and can be noticed via studying the spatial distribution of the FWI and its related codes/indices throughout the study area. For both the historic period (2010–2014) and the two simulated ones (2041–2060, 2079–2098), it can be observed that high mean values of the index as well as high numbers of days exceeding the defined thresholds tend to be concentrated in pixels located in the south and east of Greece. This can be traced to the effect that the Greek climate and topography influence the index. According to [12], who found similar patterns in the results, the southern and eastern parts of Greece tend to be the most dry and hot throughout the annual period and hence during the fire season. Before further investigation the FFWI along with the SPI and Dry50 indices, although not related to the calculations of the FWI system, they generally follow a similar spatial trend, with higher values being concentrated in pixels located in the southeast of Greece.

*3.2. Correlation between FWI and Moisture Codes*

The fire weather index is derived from meteorological information, and despite having been developed for Canadian forests, several studies have shown its suitability for the Mediterranean Basin [29,56].

As per the CFSR dataset outputs, by initially looking at Figures 4–6, one can claim similarities in patterns regarding the spatial distribution of the FWI and the other indices. All follow a similar pattern, presenting their highest values in the dry, hot south and eastern parts of Greece. For both the moisture and behavior indices, higher values can be attributed to conditions related to heat and drought in the region in which they are observed. Thus, high values of these indices define fire-prone conditions for the studied ecosystem despite ignition events themselves not being included in the FWI code. Table 3 shows the statistics for simulations in greater detail.

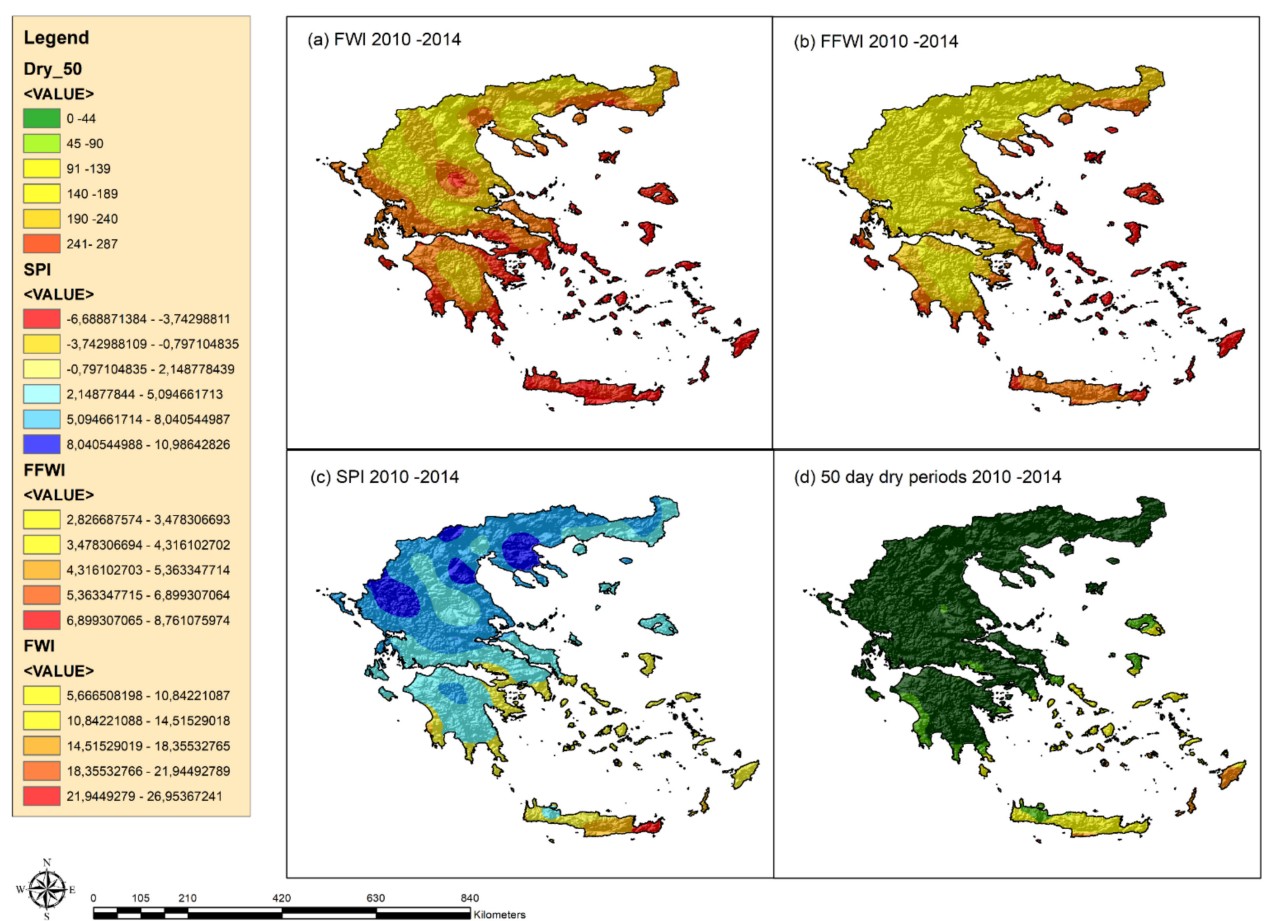

**Figure 4.** Maps for the calculated indices from the CFSR dataset for the period of May–October in 2010–2014.

**Table 3.** Simulation results for the linear regression model for the period of May–October in 2010–2014.

| Correlations | Slope | Intercept | $R^2$ | $p$-Value | Stderr. |
|---|---|---|---|---|---|
| FWI-T | 0.868 | 4.667 | 0.0305 | $1.23 \times 10^{-3}$ | $2.67 \times 10^{-1}$ |
| FWI-WS | 5.00 | 6.93 | 0.665 | $1.33 \times 10^{-157}$ | $4.342 \times 10^{-82}$ |
| FWI-Rh | −6.941695 | 19.56043 | 0.000984 | 0.565006 | 12.05175 |
| FWI-Prcp. | −6.80 | $2.40 \times 10$ | 0.840782 | $1.62 \times 10^{-136}$ | $1.61 \times 10^{-1}$ |
| FWI-FFWI | 1.42 | 4.751 | 0.880688 | $1.22 \times 10^{-157}$ | $2.85 \times 10^{-2}$ |
| FWI-SPI | −1.40 | $2.22 \times 10$ | 0.709523 | $1.76 \times 10^{-92}$ | $4.88 \times 10^{-2}$ |
| FWI-Dry 50 | $6.08 \times 10^{-2}$ | $1.43 \times 10$ | 0.537481 | $2.22 \times 10^{-58}$ | $3.07 \times 10^{-3}$ |
| FWI-Days Fwi > 40 | $3.48 \times 10^{-2}$ | 5.89 | 0.987528 | $4.64 \times 10^{-247}$ | $4.46 \times 10^{-4}$ |
| FWI-Days Fwi > 50 | $4.98 \times 10^{-2}$ | 7.87 | 0.950326 | $8.84 \times 10^{-222}$ | $6.20 \times 10^{-4}$ |
| FFWI-Prcp. | $−4.13 \times 10^{-1}$ | $1.06 \times 10$ | 0.372857 | $5.08 \times 10^{-36}$ | $9.74 \times 10^{-2}$ |
| FFWI-Rh | $1.13 \times 10^2$ | $−2.89 \times 10$ | 0.074731 | $3.18 \times 10^{-7}$ | $7.92 \times 10^{-2}$ |
| FFWI-WS | 1.86393 | 0.892362 | 0.99934 | 0.000 | 0.002608 |

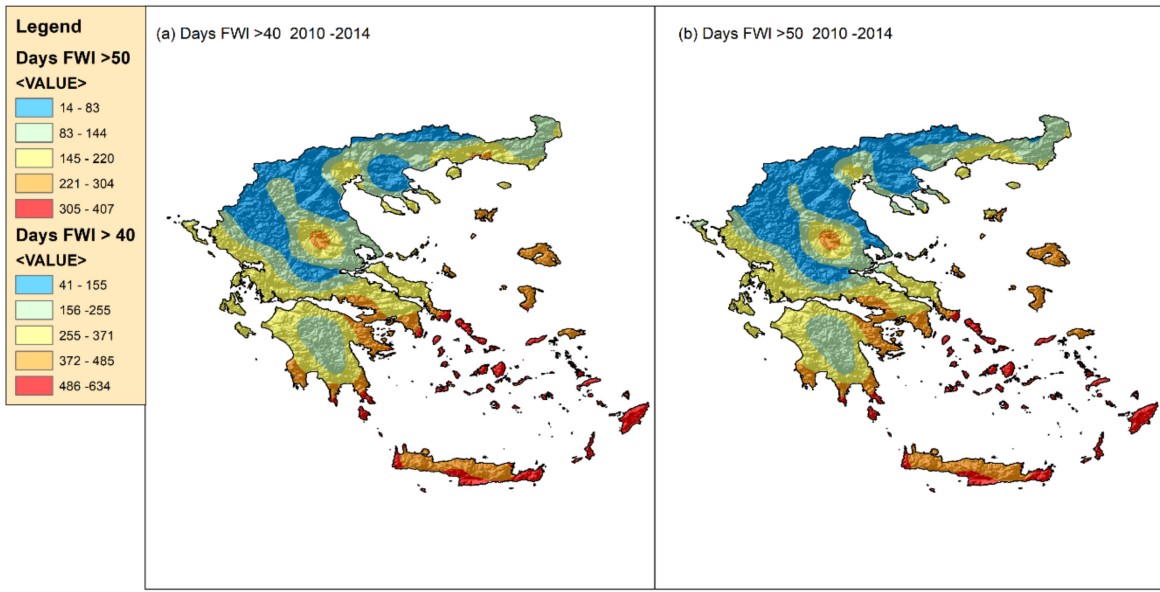

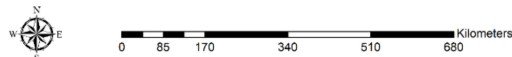

**Figure 5.** Maps for the calculated indices from the CFSR dataset for the period of May–October in 2010–2014.

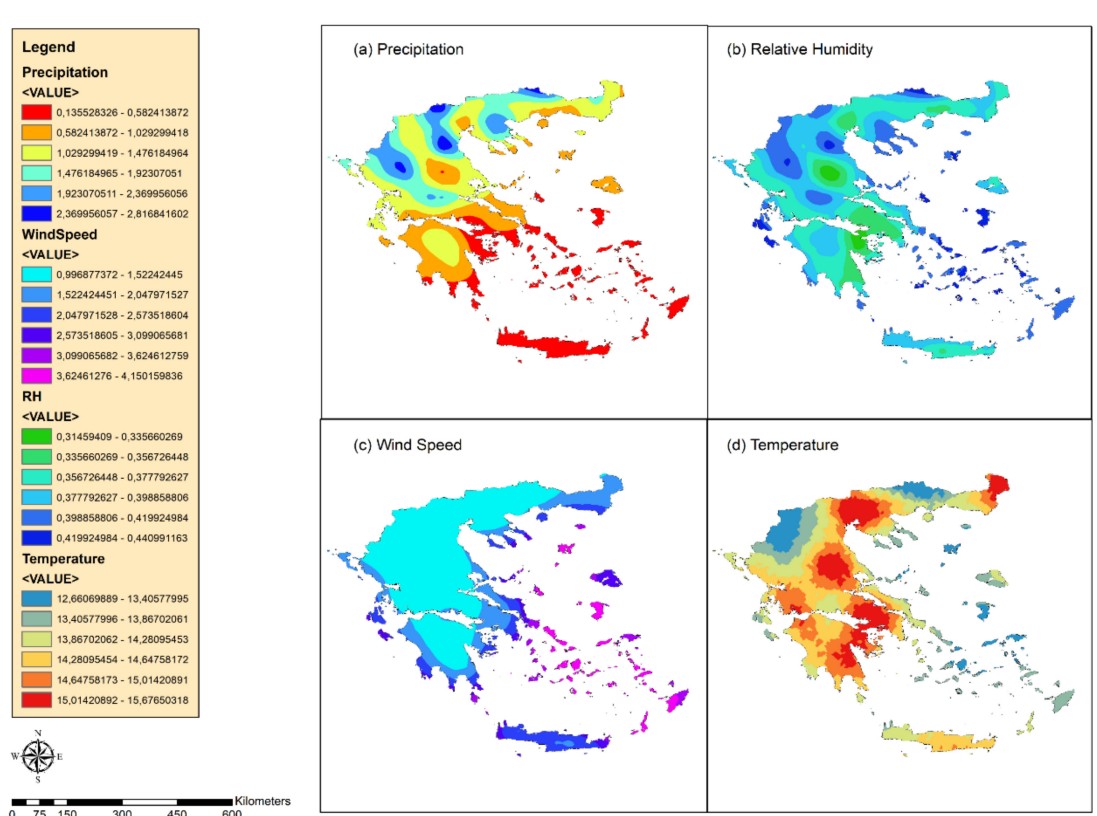

**Figure 6.** Maps for the mean values of meteorological variables of the CFSR dataset.

The model, being a machine learning program, will produce slightly different yet similar outputs each time it is run without altering its parameters. For this very reason,

the model was run in a 100-iteration scheme that extracted mean values for each output parameter of Table 3 for the total number of iterations. By examining the results of Table 3, it can be observed that the FWI correlates well with the two indices SPI and FFWI: $R^2_{SPI}$ = 0.71, $P < 0.01$ and R $R^2_{FFWI}$ = 0.88, $P < 0.01$, with the correlation between the FWI and the number of 50-day dry periods being weaker: $R^2_{Dry50}$ = 0.55, $P < 0.01$. When examining the meteorological components of the FWI, the best correlations seem to be achieved for the wind speed $R^2_{WS}$ = 0.665, $P < 0.01$ and precipitation $R^2_{Prcp}$ = 0.84, $P < 0.01$, the latter coinciding with the results of Karali et al. [12], who also found FWI and precipitation to be significantly correlated. Carillo et al. [57] also found temperature and wind speed to have positive effects on the FWI, with negative effects being attributed to precipitation and relative humidity. Temperature was found to be the most influential predictor; however, it is expected to lose significance over precipitation in future periods in the study region of the Canary Islands [57]. Correlations for temperature and relative humidity were not significant at all, producing correlation coefficients of $R^2_{Temp}$ = 0.03 and $R^2_{Rh}$ = 0.000984, $P < 0.01$. Between all of the indices, days FWI > 40 and days FWI > 50 achieve the best correlation coefficients $R^2$ throughout all four runs, all being above 0.95, $R^2_{DaysFWI>40}$ = 0.988 and $R^2_{DaysFWI>50}$ = 0.9503, $P < 0.01$. This means that concerning the CFSR dataset, there exists a strong linear proportionality between the mean values of the FWI and the number of days. It exceeds the set thresholds of 40 and 50 for the fire seasons of the study period.

The mentioned claim can also be backed by the spatial patterns that present similarity, as mentioned earlier, with the most vulnerable regions presenting high values for these three indices being mostly located in the south and east of Greece.

Although low ($R^2_{NDVI}$ = 0.45), the correlation between the FWI and NDVI presented a negative slope trendline, indicating an inhibitory effect of the healthy canopy in wildfire-favoring meteorology. A different aspect of the NDVI is also of high importance, and that is the difference between the non-fire and fire season NDVI, for which high values indicate increased biomass production during the non-fire season, a considerable percentage of which would have lost moisture or become dead biomass, acting as fuel for potential wildfires after their initiation. Such areas might be characterized as vulnerable regardless of their meteorology and hence their FWI values and would be deemed as necessary for the application of fuel treatment projects [58,59].

The model was also run for the FFWI and its three components (temperature, relative humidity, and precipitation), out of which the highest correlation was found for the wind speed $R^2_{ws}$ = 0.999, $P < 0.01$. The entire process of FWI calculation and ML linear regression was repeated for the CFSR dataset for a new time period, this time starting in March, two months prior to the typical Greek fire season. The results were more or less similar, with correlations following the same pattern and tendencies, and $R^2$ variables presenting miniscule differences in the third decimal point (Figure 7). The exception was that of the FFWI, for which the $R^2$ variable fell from 0.88 to 0.65, and the temperature, despite tripling its $R^2$, still maintained a low correlation coefficient. Tables 3 and 4 show the mean results for the 100 simulations of the two different time periods for the CFSR dataset.

**Table 4.** Simulation results for the linear regression model for the period of March–October in 2010–2014.

| Correlations | Slope | Intercept | $R^2$ | *p*-Value | Stderr. |
|---|---|---|---|---|---|
| FWI-T | 1.49 | −4.29 | $9.13 \times 10^{-2}$ | $1.39 \times 10^{-8}$ | $2.57 \times 10^{-1}$ |
| FWI-Ws | 4.63 | 7.06 | $6.63 \times 10^{-1}$ | $1.31 \times 10^{-81}$ | $1.80 \times 10^{-1}$ |
| FWI-Rh | −44.412035 | 39.008509 | 0.033759 | 0.000675 | 12.94301 |
| FWI-Prcp | −6.57 | $2.71 \times 10$ | $8.27 \times 10^{-1}$ | $1.62 \times 10^{-136}$ | $1.64 \times 10^{-1}$ |
| FWI-FFWI | 2.46 | 4.63 | $6.54 \times 10^{-1}$ | $2.85 \times 10^{-130}$ | $9.74 \times 10^{-2}$ |
| FWI-SPI | −1.54 | $2.44 \times 10$ | $7.13 \times 10^{-1}$ | $2.47 \times 10^{-93}$ | $5.34 \times 10^{-2}$ |
| FWI-Dray Days 50 | $6.82 \times 10^{-2}$ | $1.56 \times 10$ | $5.58 \times 10^{-1}$ | $9.51 \times 10^{-62}$ | $3.30 \times 10^{-3}$ |
| FWI-Days > 40 | $3.60 \times 10^{-2}$ | 6.88 | $9.84 \times 10^{-1}$ | $4.69 \times 10^{-303}$ | $2.53 \times 10^{-4}$ |
| FWI-Days > 50 | $5.38 \times 10^{-2}$ | 8.75 | $9.50 \times 10^{-1}$ | $6.05 \times 10^{-221}$ | $6.74 \times 10^{-4}$ |

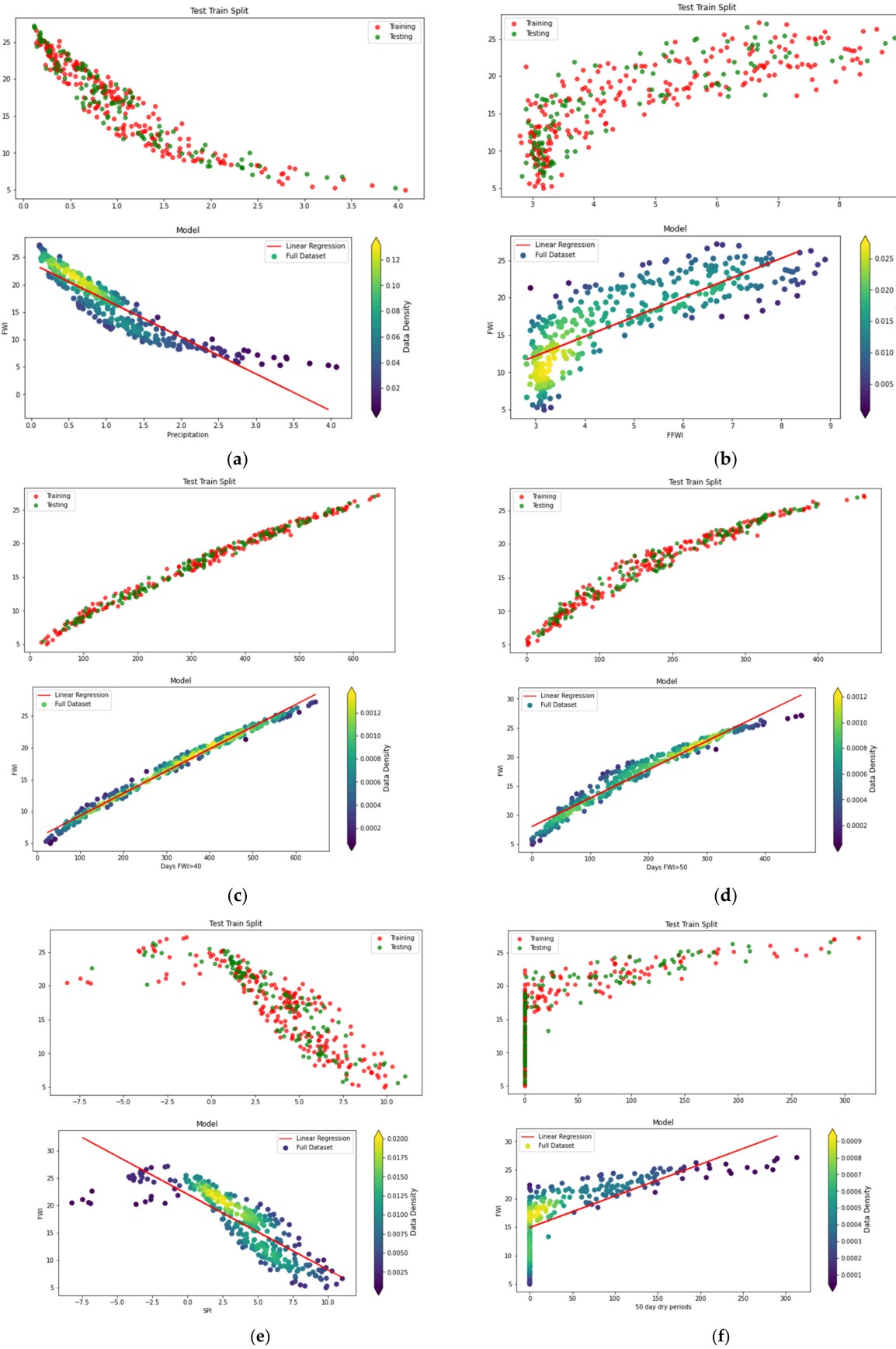

**Figure 7.** Graphic representation of the linear regression process for each index of the Swat dataset.

Sample splits are presented in top diagrams, whereas the linear fit is at the bottom of each index, (**a**) Precipitation-FWI, (**b**) FFWI-FWI, (**c**) Days FWI > 40-FWI, (**d**) Days FWI > 50-FWI, (**e**) SPI-FWI, (**f**) Dry50-FWI.

### 3.3. Influence of Meteorological Variables on FWI on a Daily/Single Point Basis

On a point basis, the strongest positive effects can be attributed to the wind speed, with temperature then exhibiting an exponential sensitivity relationship with an $R^2 = 0.996$, $P < 0.05$ for both variables. Linear increases in those components lead to exponential increases in the FWI values. Relative humidity has a linear sensitivity correlation with FWI $R^2 = 0.996$, $P < 0.05$ and presents a negative effect on the index's values. The optimal fit for the effect of precipitation was a second-order polynomial equation $R^2 = 0.885$, $P < 0.05$. Results from previous studies have shown that the FWI presents a monotonic increase/decrease with respect to its predictors [60–62]. Due to the fact that the FWI reached a near-zero (FWI < 0.5) value for a quarter of the max precipitation, for which the said modeled point was at 13.1 mm, the predictor of precipitation was kept at zero each time another variable's influence was being examined.

For the mean conditions, the index seems to increase due to incremental effects of the FWI calculations until the precipitation value of 0.5 mm, for which the FWI is 4.58, after which it begins to drop and reaches a near-zero value on the 14th day, on which there was 5 mm of precipitation (Figure 8). The same effect is observed for the maximum conditions for which the FWI peaks at 20.61 up until 0.5 mm of precipitation and reaches a near-zero value during the 16th day for a precipitation value of 7 mm.

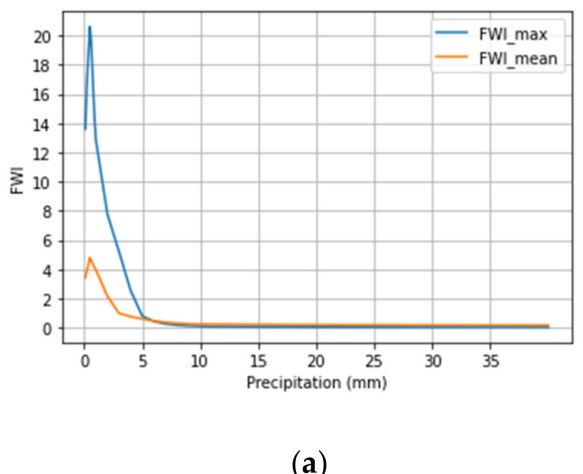

(**a**)

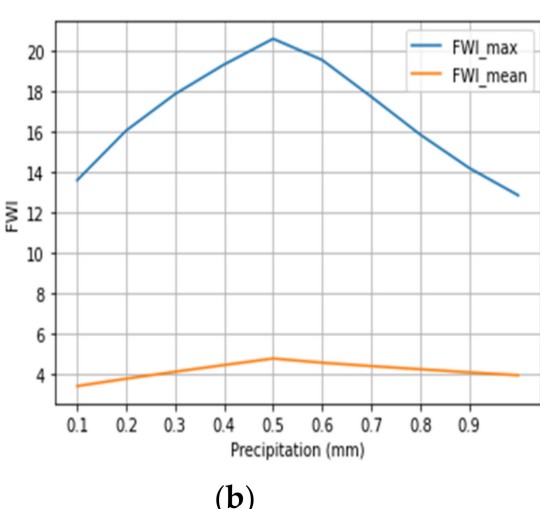

(**b**)

**Figure 8.** Effects of precipitation on the FWI values for different daily artificial precipitation values. The blue line corresponds to the maximum conditions (maximum temperature, maximum windspeed, mean relative humidity) for the studied point, while the orange line corresponds to the mean conditions (mean temperature, mean windspeed, mean relative humidity). Graph (**b**) depicts the part of graph (**a**) for the precipitation range of 0.1–1 mm in more detail. For both sets of conditions, the predictor "precipitation" presents an effect above the value of 0.5 mm.

According to the experiments of single precipitation events, the FWI is only affected for precipitation values ≥ 0.5 mm. The decrease in FWI values as well as the number of days needed for the value to recover is proportional to the amount of precipitation, with the final two values (20, 40 mm) leading to an FWI of near zero after their incidence. For the second iteration of the experiment, in which the precipitation amount was distributed across two consecutive days, the value of 1 mm showed no effects and was therefore excluded from the next iteration. The value of 5 mm decreased the FWI by a lesser amount but maintained the number of days the index needed to recover. For all other values, the end value of the FWI was less than that of the first iteration, but the number days needed

for the recovery of the index increased, with only the value of 40 mm achieving a near-zero FWI. The results in the third iteration followed the same pattern from the value of 7 mm and onward, with the value of 40 mm once again being the only one to drop the FWI to a near-zero value. For the fourth and final iteration the value of 40 mm was the only one to increase the days needed for recovery. Table 5 shows the results of this experiment in greater detail.

**Table 5.** Results for the influence of precipitation events on the values of the FWI.

| Single Day Events (15th Day) | FWI before Event | FWI after Event | Precipitation (mm) | No. of Days to Recover |
|---|---|---|---|---|
| | 31.3 | 19.24 | 0.5 | 2 |
| | 31.3 | 19.24 | 1 | 4 |
| | 31.3 | 4.66 | 5 | 6 |
| | 31.3 | 2.74 | 7 | 6 |
| | 31.3 | 1.02 | 10 | 6 |
| | 31.3 | 0.45 | 20 | 8 |
| | 31.3 | 0.26 | 40 | 9 |
| **Distributed across 2 days (15th–16th Day)** | | | | |
| | 31.3 | 8.73 | 5 | 6 |
| | 31.3 | 6.84 | 7 | 8 |
| | 31.3 | 4.66 | 10 | 9 |
| | 31.3 | 1.02 | 20 | 10 |
| | 31.3 | 0.45 | 40 | 11 |
| **Distributed across 3 days (15th–17th Day)** | | | | |
| | 31.3 | 11.75 | 5 | 6 |
| | 31.3 | 7.6 | 7 | 6 |
| | 31.3 | 7.11 | 10 | 8 |
| | 31.3 | 3.00 | 20 | 12 |
| | 31.3 | 0.69 | 40 | 12 |
| **Distributed across 4 days (15th–18th Day)** | | | | |
| | 31.3 | 1.02 | 40 | 14 |

## 4. Validation

NASA-derived datasets for the FWI were in accordance with the results from the CFSR dataset. These datasets were selected to span over the period of March–October, two months prior to the initiation of the Greek fire season, and although they have a coarser spatial resolution at $0.5°$ latitude (~50 km) by $2/3°$ (~67 km) longitude, the aim of their implementation was to validate the results and trends of the FWI on a different dataset. To capture the Greek spatial domain in the former datasets, the coordinates were specified as a bounding rectangle of latitude = (33, 43) and longitude = (19, 28). The netCDF files containing information about the FWI index and its component subindices came in a monthly format for each corresponding time period and were merged in order to capture the entire period (March–October) in each dataset. Although the time periods are not identical, the results are quite similar regarding the average values of the index for the fire seasons of each time period, and they exhibit similar spatial patterns, concentrating their higher values in the southeast of the country. Apart from studying the maps of each index for the study periods, Figures 9 and 10 capture the spatiotemporal variation in the indices of the FWI system in response to geographic latitude. Such charts were also produced for all NASA datasets. The results showed similar patterns demonstrating the descending tendencies of the FWI and its components, with increasing latitude within

the Greek domain. The tendencies and patterns for all of the NASA datasets (GPM-Final, GPCP, TRMM long-term mean) are similar.

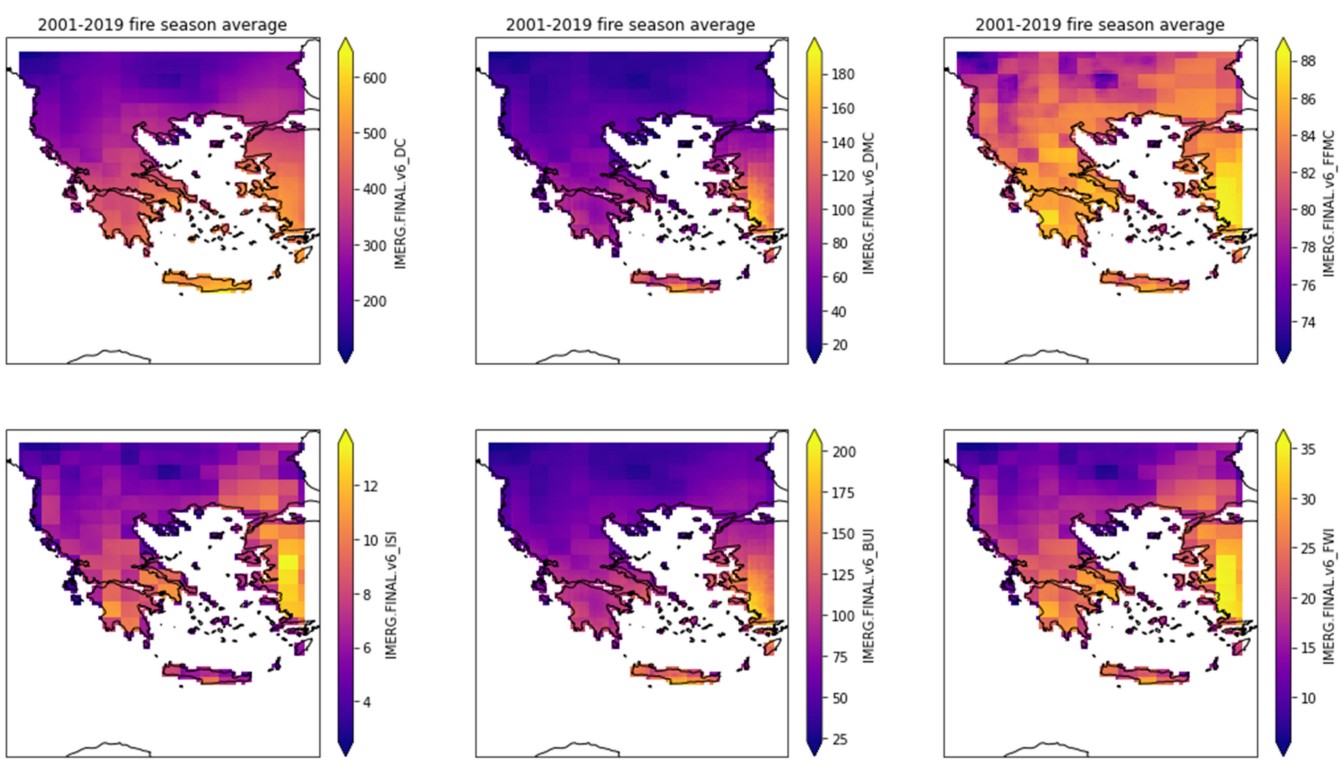

**Figure 9.** FWI system indices for Immerg.v6 dataset for 2001–2019 fire seasons (March–October) for a bounding box (lat = [33, 43], lon = [19, 28]).

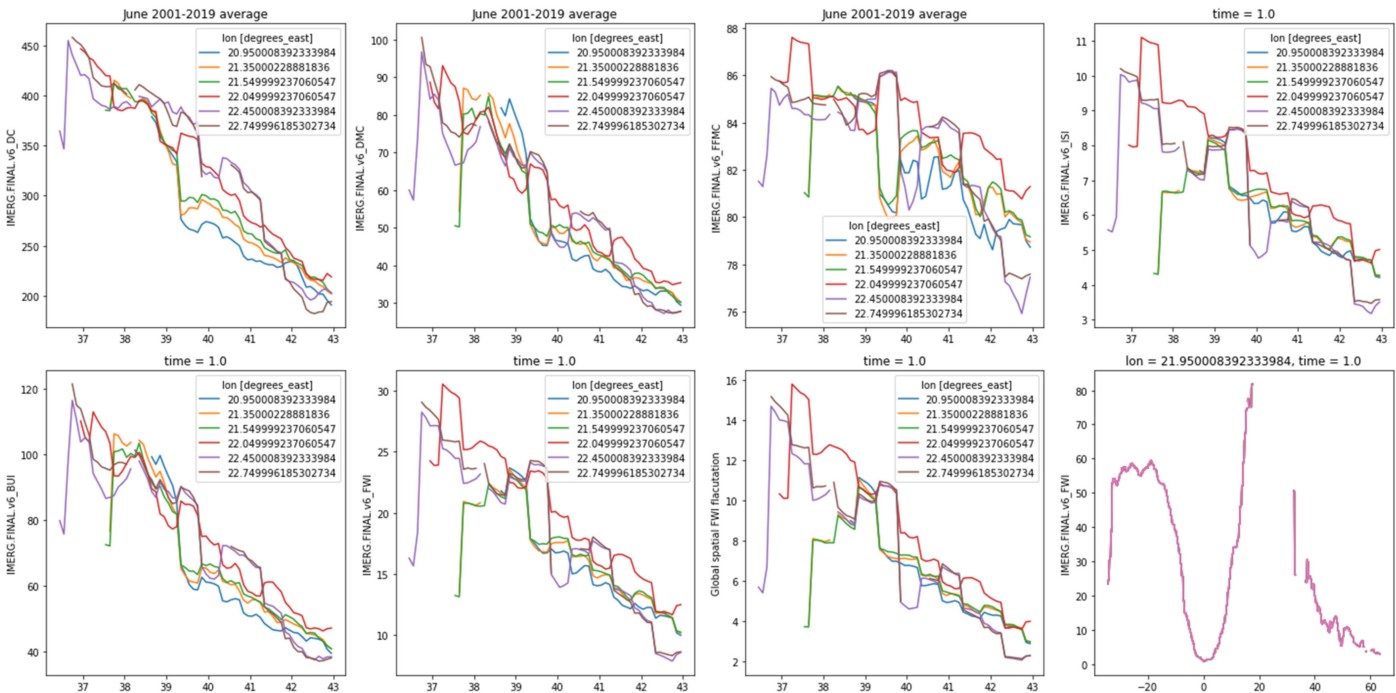

**Figure 10.** FWI system indices with respect to latitude for fixed longitudes (colored lines) for a bounding box (lat = [33, 43], lon = [19, 28]) for Immerg.v6 dataset for 2001–2019 fire seasons (March–October). Graph after the FWI represents to the variation of the daily severity index (DSR) regarding latitude, while the final graph represents FWI variation on a global scale.

The FWI was also calculated for the eight meteorological stations mentioned in Section 2.2. The index was calculated on a daily basis for the period of 2010–2020, only excluding the months of December and January. Similarities in the patterns seem to appear between the FWI timeseries and those of the meteorological components with either positive or negative relationships between values. The linear regression model was again implemented, but within the scale of a 10-year time series, the results showed very poor linear correlation. A Poisson regression model was chosen to be implemented to study the influence of the components of the FWI on its values for each one of the stations. The model uses the maximum likelihood estimation (MLE) method in Equation (5).

$$P(y|x) = \frac{e^{-\lambda} * \lambda^y}{y!} \tag{5}$$

$$\lambda = e^{x\beta} \tag{6}$$

Variables $x$, $y$, $\lambda$, and $\beta$ are vectors, where $x$ is the predictors (independent variables), in this case of the meteorological components, and y is the values of the FWI that are aimed to be predicted. $\beta$ represents the coefficients for each predictor variable (temperature, relative humidity, precipitation, wind speed). The probability of occurrence for all of the values of the $y$ vector (FWI) is also called joint probability and is given by Equation (7).

$$L(\beta) = P(y|X) = \frac{e^{-\lambda 1} * \lambda 1^{y1}}{y1!} * \ldots * \frac{e^{-\lambda n} * \lambda n^{yn}}{yn!} \tag{7}$$

where $n$ is the length of the vectors. Taking the logarithm of Equation (7) will give the log's likelihood function, which is simpler to derive. Solving $\frac{dL}{d\beta}$ will give the maximum likelihood estimate $\beta$ vector containing the $\beta 1 \ldots \beta n$ coefficients. For the Poisson regression model, the dataset was split following a 0.8/0.2 train/test ratio. Model accuracy was satisfactory for all stations, with $R^2$ being higher than 0.7, and pseudo $R^2$ was above 0.99 for all stations. Higher $\beta$ coefficient values indicate stronger influence of the component predictor in the value of the FWI. Precipitation was found to be the best-performing predictor, with coefficients being one order of magnitude higher than the rest of the components, with $\beta_{precipitation}$ ranging between $-0.179$ and $-0.3845$, with the negative connotation indicating an inverse relationship between values. The next best-performing predictor was the wind speed, which had a positive connotation. The aforementioned results are more thoroughly shown in Table 6; Table 7 is in accordance with previous results from the study regarding the correlation of FWI and its components as well as results from [12] despite the non-linear trends. Figure 11 depicts the predicted FWI series (green) versus the actual calculated FWI series (red) for the Poisson regression model for the station in Volos.

**Table 6.** Results from the Poisson regression model for the 8 stations in the "CLIMPACT" network for the period of February–November in 2010–2020.

|                | $\beta1$ | $\beta2$ | $\beta3$ | $\beta4$ | Intercept | Pseudo $R^2$ | $R^2$ |
|----------------|----------|----------|----------|----------|-----------|--------------|-------|
| Athens         | 0.0539   | −0.0226  | 0.0623   | −0.3695  | 1.698     | 0.997        | 0.746513 |
| Drama          | 0.0451   | −0.0293  | 0.0808   | −0.495   | 1.9016    | 0.9962       | 0.808201 |
| Nevrokopi      | 0.0449   | −0.0314  | 0.1023   | −0.418   | 1.9137    | 0.9947       | 0.791922 |
| Rethymno       | 0.0597   | −0.015   | 0.041    | −0.1791  | 1.3334    | 0.9692       | 0.7644 |
| Alexandroupolis| 0.0527   | −0.0229  | 0.0512   | −0.2463  | 1.6568    | 0.999        | 0.773696 |
| Florina        | 0.0512   | −0.0307  | 0.0929   | −0.3845  | 1.8308    | 0.9958       | 0.80982 |
| Volos          | 0.0583   | −0.0234  | 0.0579   | −0.3624  | 1.4501    | 0.993        | 0.776161 |

**Table 7.** *P*-values for the Poisson regression model.

| Stations | p1 | p2 | p3 | p4 |
|---|---|---|---|---|
| Athens | 0.0 | 0.0 | 0.0 | 0.0 |
| Drama | 0.0 | 0.0 | 0.0 | 0.0 |
| Nevrokopi | 0.001 | 0.001 | 0.005 | 0.015 |
| Rethymno | 0.01 | 0.0 | 0.01 | 0.07 |
| Alexandroupolis | 0.01 | 0.01 | 0.01 | 0.01 |
| Florina | 0.0 | 0.0 | 0.0 | 0.0 |
| Volos | 0.0 | 0.0 | 0.0 | 0.0 |

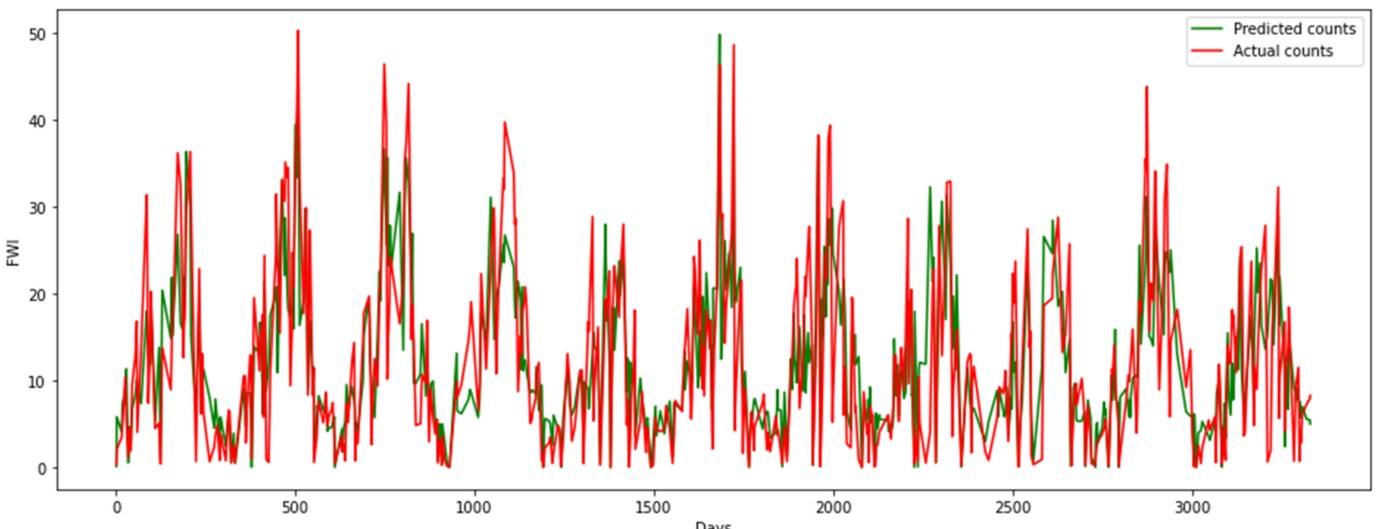

**Figure 11.** Predicted FWI series (green) versus actual calculated FWI series (red) for the Poisson regression model for the station in Volos. Number of days is displayed on the *x* axis for the time period of February–November) in 2010–2020, excluding the days with no meteorological counts.

## 5. Future Projections

When it comes to future projections of the index, we note increasing trends regarding the mean and max values of the FWI in both time periods for both RCPs. Specifically, the mean value for the CFSR dataset covering the historic period of 2010–2014 is 16.84, while the mean values for the index of RCP 4.5 are 36.94 for the first period (2041–2060) and 38.19 for the second (2079–2098). For RCP 8.5, the values are 38.24 and 43.76, respectively (Table 8). The max FWI for the CFSR dataset was 27.14, with the max future values of 88.55 and 89.73 for the two periods, respectively, for RCP 4.5 and 91.37 and 99.84 for RCP 8.5. As per the days exceeding thresholds, the numbers tend to increase for all three fire danger classes: low, medium, and high, not only as we move from the first to the second time period but also by comparing RCP 4.5 (Figure 12) with the pessimistic RCP 8.5 (Figure 13). The indices days FWI > 40 and days FWI > 50, which are dedicated to the CFSR dataset covering the historic period, refer to the fire seasons of the five-year period; hence, it is advised that the number be divided by 5 (the number of years in the period) when examining the data on an annual basis. Comparing historic and future states, the number of days exceeding thresholds seems to increase when examining RCP 8.5, especially in the latter period of 2079–2098. A common trend regarding the current and future projections in all periods and RCPs that transcends the dataset barrier is the tendency of high FWI values and high numbers of threshold-exceeding days concentrated in the drier and hotter southern and eastern parts of Greece, and, more specifically, the vulnerable areas seem to be central Macedonia, Thessaly, Attica, and Crete, a trend also persistent in historic period simulations.

**Table 8.** FWI mean and max values for datasets and their corresponding study periods.

| Dataset | Mean FWI | Max FWI | Std. Dev. |
| --- | --- | --- | --- |
| RCP 4.5 2041–2060 | 36.94 | 88.55 | 19.48 |
| RCP 4.5 2079–2098 | 38.19 | 89.73 | 19.75 |
| RCP 8.5 2041–2060 | 38.24 | 91.37 | 19.72 |
| RCP 4.5 2079–2098 | 43.76 | 99.84 | 19.87 |
| CFSR dataset 2010–2014 | 18.49 | 30.56 | 6.159 |

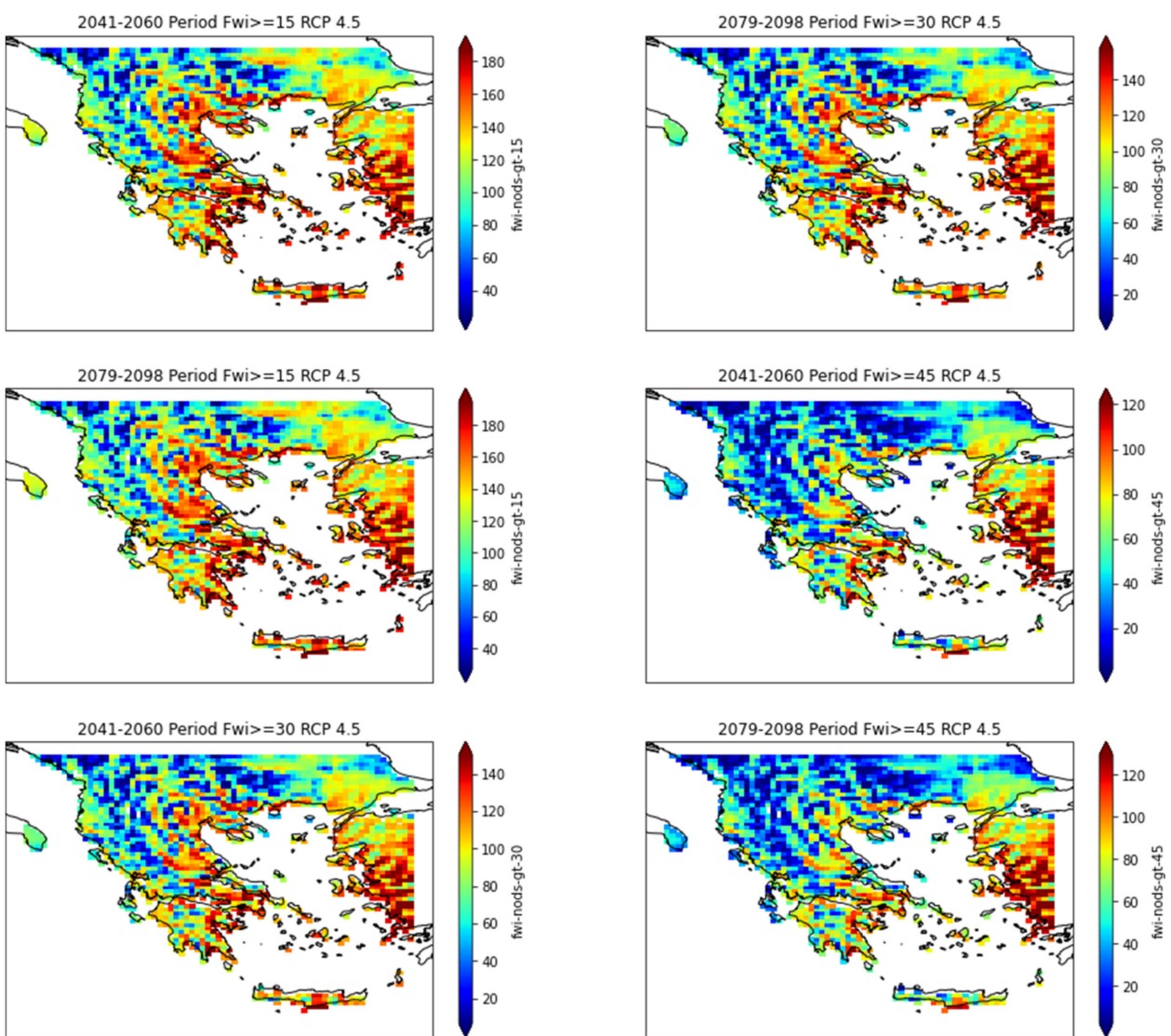

**Figure 12.** Graphic representation of the number of days per pixel the FWI exceeds the thresholds of 15, 30, and 45 defining the three fire danger classes: low, medium, and high, for the full time periods 2041–2060 as well as for 2079–2098 regarding the mean case of the RCP 4.5.

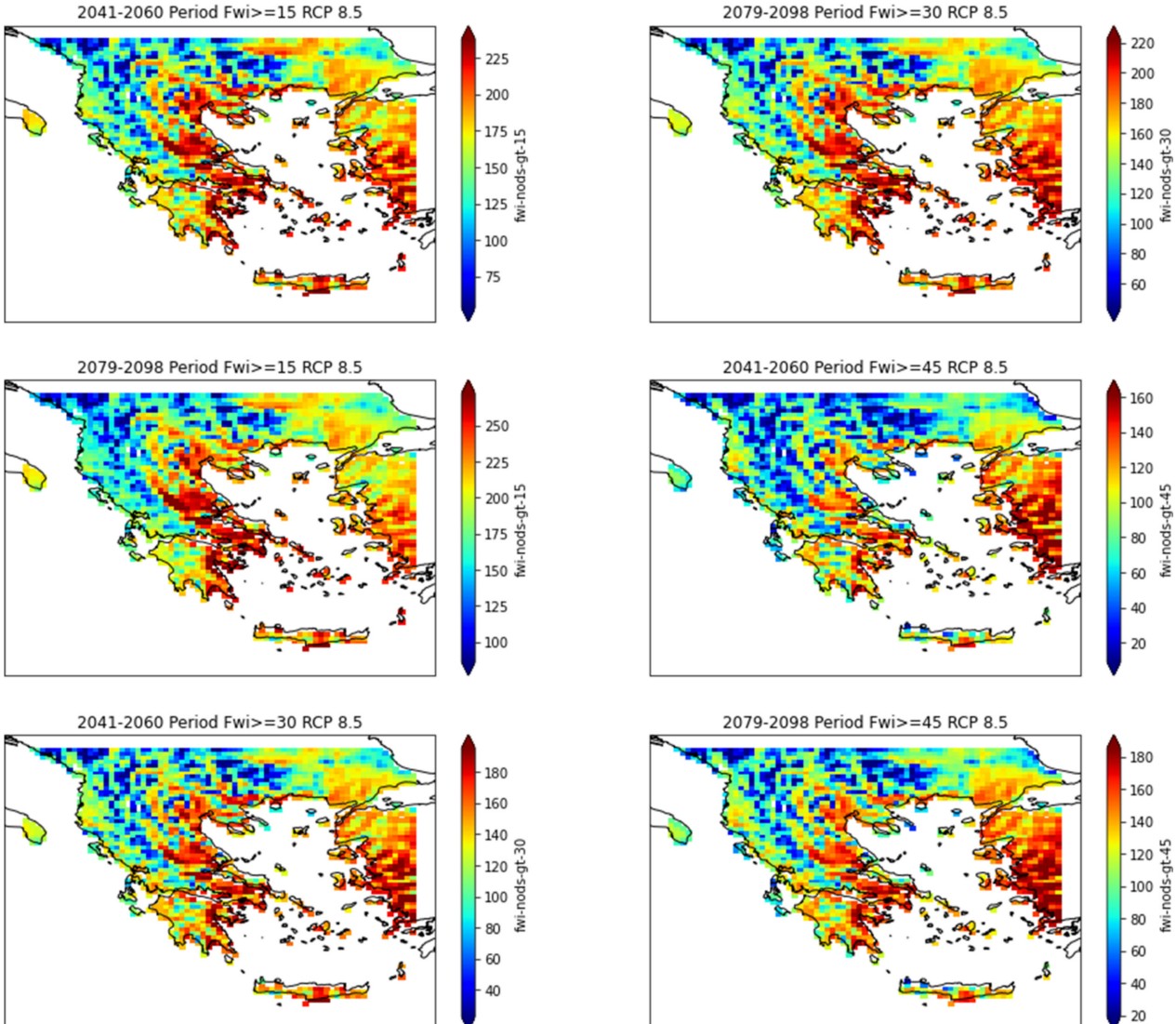

**Figure 13.** Graphic representation for the number of days per pixel the FWI exceeds the thresholds of 15, 30, and 45 defining the three fire danger classes: low, medium, and high, for the full time periods of 2041–2060 as well as 2079–2098 regarding the mean case of RCP 8.5.

The results of the Mann–Kendall test show no trends when the FWI time series is examined under the RCP 4.5 scenario, with the exception being the points with the coordinates rlon = (3,4) and rlat = −5 (Figure 14). Increasing trends were found for all pairs of points when examining the RCP 8.5 scenario Figure 14). *P*-values fall well below the 0.05 threshold for RCP 8.5, with *P* > 0.05 for all points for RCP 4.5, except for those with increasing trends. Strong correlations (Kendall τ > 0.3) only appear for RCP 8.5, for the same RCP z-scores are also higher, reaching up to z > 4, indicating further deviations from the series mean. The S-values are also higher when examining RCP 8.5, indicating a higher accumulation of positive differences as one iterates over the time series values. Finally, the slopes of the trendlines are also higher for RCP 8.5, indicating a higher gradient for the FWI values for the total merged period of 2041–2098.

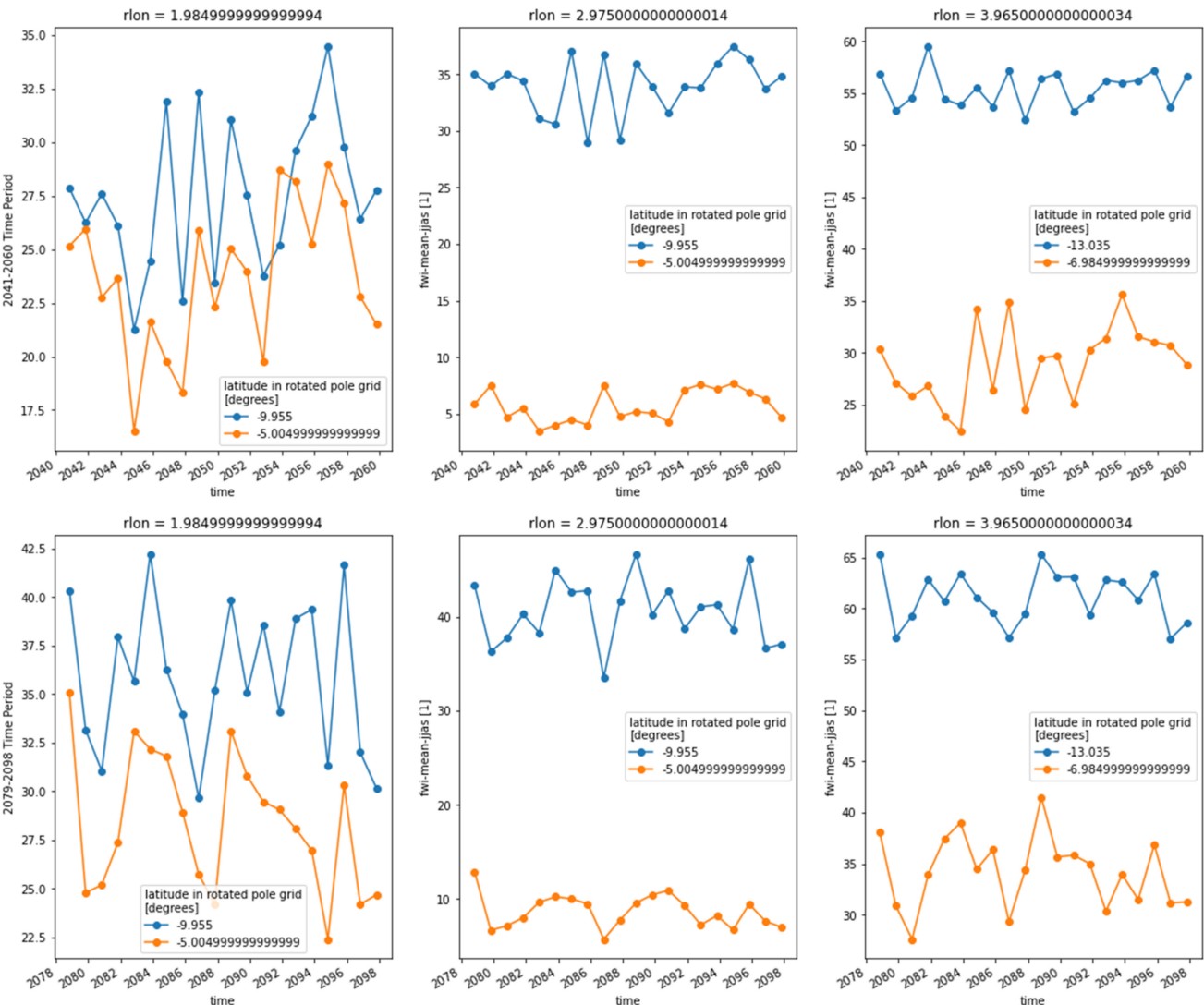

**Figure 14.** Seasonal variation in FWI for sets of latitudes (colored lines) for fixed longitudes (top of each diagram). The diagrams correspond to RCP 8.5 for the periods 2041–2060 (**top** row) and 2079–2098 (**bottom** row). Coordinates are in rotated North Pole coordinate system, where lower values correspond to closer equator latitudes; hence, the blue lines have a greater range than the FWI values.

Table 9 shows percentage differences in the z-score, slope, Kendall-τ, and S, points that exhibit no trends for RCP 4.5. Increasing trends in RCP 8.5 were given the codename "01", whereas points that presented increasing trend for both RCPs were codenamed as "11". For the variables of z-score, Kendall-τ, and S, most points present an increase of over 100%, with point p3 presenting a near 200% increase. These increases signify higher FWI values as the time series moves further in the time axis as well as higher deviations from the series mean. The same can be said for point p5, with the only difference being that the increase is nearly 500% for all three of the mentioned values. Increases in slope ranging between 170 and 290% indicate increases in the gradient, with p5 presenting an increase higher than 900%. Point p6 is an outlier compared to the rest, presenting minor increases in the z-score, τ, and S (nearly 24%) and an increase in slope of 128%. This can be attributed to the fact that the point presented an increasing trend for both RCPs; hence, differentiation between the two scenarios was lower.

**Table 9.** Percentage differences of Mann–Kendall test values.

| Points | p1 | p2 | p3 | p4 | p5 | p6 |
|---|---|---|---|---|---|---|
| Trends for RCPs | 01 | 01 | 01 | 01 | 01 | 11 |
| z-score | 1.437919 | 1.242823 | 2.1219 | 1.018425 | 4.982456 | 0.238683 |
| Kendall-τ | 1.428644 | 1.230825 | 2.103448 | 1.012195 | 4.896552 | 0.237705 |
| S | 1.428571 | 1.230769 | 2.103448 | 1.012195 | 4.896552 | 0.237705 |
| slope | 2.609017 | 1.725341 | 2.928356 | 2.117479 | 9.162352 | 1.286072 |

Climate models are highly valuable tools in the domain of climate change and in climate science in general. They can provide useful information that would otherwise be impossible to obtain. However, as mentioned by Karali et al. [12], climate models are unavoidably accompanied by errors that are a function of many factors, such as the model itself. As the mainly empirical calibration and the simulation period increase the error, the further it strays away from the current one. Hence, the reader and, to a higher extent, the user should be aware of that uncertainty when a study requires the application of such models.

## 6. Discussion

The results of the FWI values present a higher dependence on wind speed. Precipitation, while not significant in Iberia, presented higher skill as a predictor variable in the region of Greece [6]. Moisture content in the fuel layers is the most significant parameter affecting ignition and fire propagation, and it is hence reasonable to expect that it should be directly related to forest fire risk and behavior [63], which are reflected in the values of the FWI. Dimitrakopoulos et al. [30] found fire occurrence to be highly correlated with the DC and DMC components of the FWI ($r_{DC} = 0.89$, $r_{DMC} = 0.78$). Karali et al. [12] found the areas most affected by climate change to be located in the southeast, particularly in central Macedonia, Thessaly, Attica, and Crete. According to Varela et al. [29], the burned area and number of fires show a strong correlation with FWI percentile indices, which seem to be exponential. Taking into account results from Karali et al. [12], a very high number of extreme days were found for several regions, mainly in the southeast of Greece, where forest fires seem to be present in an increased concentration. The ML model results for the FFWI come in accordance with results of Kambezidis et al. [64], who also found wind speed to be the best-performing component regarding correlation with the FFWI.

Jolly et al. [65] found increasing trends in the mean amount of rain-free days for current-period datasets for 1979–2013. Giannakopoulos et al. [66] examined changes in the Mediterranean basin under a 2 °C warming scenario. Heatwave days are projected to increase by up to 1 month in continental regions and by 2 weeks in coastal ones. Precipitation is expected to present with a drop within the scale of 10–20% annually, with projections for the period of 2031–2060 dictating an increase in the total annual dry days of up to a week for coastal areas and up to three weeks for continental ones. Increasing trends in temperature and meteorological drought are expected to increase both the mean values of the FWI as well as the days where it exceeds the very high and extreme thresholds, negatively affecting fuel moisture and thus promoting fire occurrence and spread.

According to Jolly et al. [65], fire season length increases tend to be influenced not only temperature but also by changes in humidity and the length of rain-free intervals as well as by wind speed. Results from Julien et al. [67] show that wildfires tend to grow larger when short-term fire weather conditions are combined with long-term summer drought. Julien et al. [67] also proved that the most extreme wildfire events in the Mediterranean Basin mostly occur during fire regimes characterized by the combination of heatwaves and prolonged drought conditions, further backing the influence of precipitation in fire weather and hence the favorability of fire occurrence.

## 7. Limitations and Future Direction

Data from the CFSR model only reach as far as 2014. The mentioned data were modeled outputs; hence, there is an expected degree of uncertainty errors compared to actual values. The Climpact dataset could compensate for such shortcomings, though the number of available stations is still limited, and some of them have null observations for particular dates, though interpolation techniques could be used to fill null observations. NASA datasets have a very low spatial resolution to accurately represent patterns and variations in small and topographically heavy domains. Future FWI studies come with an expected error as well due to model uncertainty. The biggest overall limiting factor was data access and availability for the Greek domain, which was the main reason for selecting the mentioned data sources. Further data availability of the modeled and observed datasets regarding FWI series as well as meteorological values for computation could lead to a more sufficient validation of the results for the study area. The FWI is calculated with values observed at 12 p.m. The absence of such observations is another limiting factor affecting the accuracy of predictions. This was compensated for by the selection of the maximum temperature and minimum relative humidity when these values were provided by the datasets. In order for the data availability problem to be solved, future efforts will be aimed at calculating and studying the FWI in GEE using available meteorological datasets. The use of the GEE will also eliminate errors related to interpolation methods since data coverage is constant for the area encompassed by the datasets. Finally, as the FWI itself does not predict ignition but rather weather favorability over it, further factors would need to be studied if fire occurrence is a main concern (fuel properties, soil moisture, land use, population density, etc.).

## 8. Conclusions

In general, the results show that the southern and eastern drier parts of Greece mostly accumulate the highest values of the FWI as well as higher values revealing drought and fire favorability regarding the other indices (SPI, Dry50, and FFWI. At the same time, they accumulate high numbers of days where the FWI exceeds the values of the "Very High" and "Extreme" thresholds. NASA datasets are in accordance with the CFSR-derived FWI dataset regarding spatial distribution despite coming from a different dataset.

The linear regression process in Section 3 proved that high numbers of mean FWI signify a high number of threshold-exceeding days in an area. There also seems to be a strong correlation between the FWI and the wind speed, which is also backed by the strong correlation between the FWI and the FFWI. The SPI index, while achieving a weaker correlation compared to FFWI, still maintains a strong link to the FWI values, with $R^2 = 0.71$. The best correlation-achieving component of the FWI was that of precipitation: $R^2 = 0.88$, $P < 0.01$. Precipitation was also found to be the most influential factor-predictor in the Poisson regression model for the eight meteorological stations. The results maintain similar tendencies for both the typical Greek fire season (May–October) as well as the extended one (March–October). *P*-values for all simulations are way below the 0.01 threshold, leading to the exclusion of the claim of random chance affecting and defining the results.

The influence of precipitation on the values of the index could be attributed to the strong influence in the three moisture indices: FFMC, DMC, and DC. All subindices were found to present a high correlation with the FWI, with $R^2$ values of 0.87, 0.82, and 0.87, respectively, and their associated slope coefficients were −4.3, −8.15, and −10.1. The FFMC expressing moisture content in the top fine layers of the ground has a response time to meteorological changes of only 16 h, with the DMC having one of 10–12 days. Quick response accompanied with a high negative correlation explains the influence on the values of the FWI. Giannaros et al. [68] also found the FFMC to have the highest weight regarding FWI sensitivity

The results presented in Section 3.3 reveal the influence of precipitation on the values of the FWI. The effect of precipitation becomes apparent in the FWI values above the threshold of 0.5 mm, which is enough to drop the index's value to a lesser fire danger

category. The decrease is proportional to the amount of mm of precipitation, with the same being true for the number of days the index needs to be restored to close to its prior value. The results also showed that the index needs more days to recover when the said precipitation amount is distributed across consecutive days rather than being concentrated as a single-day event for as long as the daily values remain above the 0.5 mm threshold. The results, however, could be downplayed or amplified by the soil type of the region, something not taken into account by the FWI calculations.

Future projections show similar patterns to current datasets, concentrating the high number of threshold-exceeding days in the south and east, which is the same area that seems to be the most affected when examining different climate change scenarios. Overall, the results show increasing trends, suggesting a two-fold increase between current and future datasets for the mean FWI values, with differences reaching up to three-fold increases regarding the maximum FWI, although model uncertainty should also be taken into account.

The aim of this study was to highlight the link between FWI and climate conditions, especially drought, among other factors that affect its spatial profile, such as topography. Generally, the drier and hotter southern and eastern regions of Greece seem to concentrate high values of FWI and have a higher numbers of fire events. The FWI alone cannot predict fire occurrence, since being exclusively derived by meteorological information, it can only provide a measure assessing the favorability of weather conditions regarding fire occurrence in a study area. Studying ignition and fire behavior specifically would necessitate the inclusion of further metrics such as fuel information (quantity, structure, moisture condition), flame length predictions, and crown bulk density, as well as historical information about ignition, fire intensity, and duration [58,59] and information about population- and human-driven factors [69]. However, as highlighted by Varela et al. [29], meteorology alone can provide a metric of fire danger classification for a region with areas with a high or extreme FWI representing the majority of fire events.

**Author Contributions:** Methodology, L.V.; Project administration, M.S.; original draft, N.N.; review & editing, N.M. All authors have read and agreed to the published version of the manuscript.

**Funding:** This research received no external funding.

**Data Availability Statement:** Data for the calculation of the FWI were obtained from (https://swat.tamu.edu/data/cfsr (accessed on 18 March 2022)) in the form of .csv SWAT input files. The data were produced by the CFSR (Climate Forecast System Reanalysis (CFSR). For future predictions the data came in the form of netCDF files provided by the European Commission for Medium Weather Forecasts (ECMWF). Meteorological data from Athens, Drama, Nevrokopi, Rethymno, Alexandroupolis, Florina, Volos were retrieved from "CLIMPACT" network (https://climpact.gr/main/ (accessed on 18 March 2022)).

**Acknowledgments:** The authors acknowledge N.R. Dalezios for the encouragement to complete this research. An initial version of this paper was presented at the 2nd International Symposium on Sustainability and Geoinformatics in Vulnerable Ecosystems (2nd AGROECOINFO), which ran from 29 June to 2 July 2022, Volos, Greece. The authors would like to thank the Guest Editor and the two anonymous reviewers for their constructive and useful comments, which contributed to an improved version of the initial paper.

**Conflicts of Interest:** The authors declare no conflict of interest.

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
