# Peer review of "Contribution to the Study of Forest Fires in Semi-Arid Regions with the Use of Canadian Fire Weather Index Application in Greece"

_climate, doi:10.3390/cli10100143_

Round 1
Reviewer 1 Report
Revision: Contribution to the study of forest fires in semi-arid regions with the use of Canadian Fire Weather Index. Application in Greece
Though the overall theme of this study is relevant and UpToDate in my opinion it lacks some details, rigor in the presentation, and statistical analysis. The lack of rigor in the nomenclature used shows some inexperience of the authors in the use of climate models. I think that the authors should revise the main objectives of this study and focus the analysis in attain them. The overall methodology should be revised in order for the results to be supported by strong statistical arguments.
I advise authors to carefully read the text and correct grammar, the English, and the multiple typos present throughout the manuscript. Namely, the text of the word template that is still present at the beginning of section 2 lines 116-122: ‘The Materials and Methods should be described with sufficient details to allow others to replicate and build on the published results. Please note that the publication of your manuscript implicates that you must make all materials, data, computer code, and protocols associated with the publication available to readers. Please disclose at the submission stage any restrictions on the availability of materials or information. New methods and protocols should be described in detail while well-established methods can be briefly described and appropriately cited.’ Authors should carefully read the author guidelines of this journal.
For example, the Authors use both Fire Weather Index (FWI) and the FFWI which is referred to as the Fosberg Fire Weather Index. Though details regarding the well-known FWi are provided, few are presented for the not-well-known FFWI. The descriptions of the indices should be similar. A more detailed description of the SPI along with the scale should also be provided. This subsection also lacks references. The same for the NDVI description. All methodology should be clearly stated in this section. It is very confusing to follow if the authors are using monthly data, daily data, which dataset, if they are using the fire season months (this should also be stated), and so on, for each section in the results. The MLE should be presented in this section and not in the results in my viewpoint.
All figures should also be revised. First, the legend presents two FWI there are 7 captions for 6 images, something is wrong (figure 2, for example). Also, the continuous scale makes the interpretation of the images very difficult since for example the color barrier for SPI between the negative and the positive values cannot be depicted. I strongly advise an interval scale for all figures.
I find it very difficult to understand what the authors intend to achieve as solid results in section 3.2. . Due to the size of the study area and the differences in orography and vegetation, besides other aspects, performing a linear correlation will not give us scientifically robust and statistically significant results. Authors should perform correlation pixel-by-pixel and shows the patterns of the statistically significant pixels. This will show you the regions in which the correlations are significant besides their strengths.
Typically, FWI is computed daily thus allowing civil protection authorities to implement the necessary measures to prevent major wildfires. Therefore, the study of the monthly variation of the FWI does bring some insight into its climatology. Let me recall that the meteorological conditions vary daily, thus impacting some of the components of the FWI, which in turn will impact the FWI itself. In section 4, the authors state: ‘The netCDF files containing in formation about the FWI index and its component sub-indices came in a monthly format for each corresponding time period and were merged to capture the entire period (March – October) in each dataset. Although time periods are not identical the results seem to be similar regarding average values of the index for the fire seasons of each time period, and they seem to follow a similar spatial pattern, concentrating their higher values in the southeast of the country.’ Seem to follow does not sound like a very scientific statement. Furthermore, the spatial distribution (pixel by pixel/coordinates) is also not coincident. Again, ‘Tendencies and patterns for all NASA datasets (GPM-Final, GPCP, TRMM long term mean) are similar.’ in lines 444/445. There are statistical techniques that ensure that the tendencies (or trends) and patterns are indeed statistically significant. This for me is a huge problem in the study presented in this manuscript. These statements are also presented in section 5 Lines 492/493 ‘When it comes to future projections of the index, we note increasing trends regarding the mean and max values of the FWI in both time periods for both RCPs.’ Authors cannot state this just by looking at the spatial patterns or by averaging FWI values for the study area and fire season. Means tend to be highly sensitive to outliers, but also to strongly smooth the values in a series. The trends should also be tested for their statistical significance.
A discussion should be provided, and the conclusions should also be readdressed since the ‘take-home’ message is not scientifically robust nor clear from my viewpoint.
Minor comments:
RCP: this acronym is not defined in the text. Also, when using it, it should be preceded by the word under, e.g. under RCP 4.5…
Line 335: it is not correct to say FWI is meteorologically derived… FWI is derived from meteorological variables or parameters…
Line 339/340: This is not scientifically correct. One can manipulate the color scales to match the patterns…
Line 422: [10] found… this does not follow the MDPI author’s guidelines. It should be written as: Karali et al. [10] found … The same comment should be applied throughout the text.
All references should be revised, there are incomplete references. Also, I advise authors to read other works regarding FWI, namely in Italy, Spain, and Portugal.
Captions: Captions should be revised, again under RCP 4.5 or 8.5. Figures 7 and 8: for me is not clear if the authors use daily values from January 1st, 2041 until December 31st, 2060, or daily values between the fire season. This should be clearly stated previously in the methodology section, but also in the captions.
Author Response
Comments from the Reviewers
Reviewer #1 - Comments
Though the overall theme of this study is relevant and UpToDate in my opinion it lacks some details, rigor in the presentation, and statistical analysis. The lack of rigor in the nomenclature used shows some inexperience of the authors in the use of climate models. I think that the authors should revise the main objectives of this study and focus the analysis in attain them. The overall methodology should be revised in order for the results to be supported by strong statistical arguments.
The authors thank the reviewer for his/her constructive and useful comments. All comments made by the reviewer have been addressed and the mistakes have been corrected in the revised paper (see revised manuscript and annotated manuscript). The reviewer comments are shown in “plain text” and the authors’ response in the text with italics. Furthermore, new scientific elements are incorporated in the manuscript and supported with statistical analysis and discussion and the revised manuscript has been completely redesigned and restructured accordingly.
- I advise authors to carefully read the text and correct grammar, the English, and the multiple typos present throughout the manuscript. Namely, the text of the word template that is still present at the beginning of section 2 lines 116-122:.
The author comments have been addressed in the revised manuscript (see revised manuscript and annotated manuscript).
- The Materials and Methods should be described with sufficient details to allow others to replicate and build on the published results. Please note that the publication of your manuscript implicates that you must make all materials, data, computer code, and protocols associated with the publication available to readers. Please disclose at the submission stage any restrictions on the availability of materials or information. New methods and protocols should be described in detail while well-established methods can be briefly described and appropriately cited.’ Authors should carefully read the author guidelines of this journal..
The authors agree with the reviewer and the comments have been addressed in the revised manuscript. Section 2 “Materials and Methods” has been completely revised and all subsections are discussed and cited accordingly. Additional information is provided in Sections 2.4 (Mann Kendall test) and 2.5 (FFWI, SPI NDVI additions with equations). New subsections have been added in the revised manuscript (2.6: Use of Machine Learning Linear Regression Model to find correlations between indices and 2.7: Examination of the influence of meteorological variables on FWI a daily /single point basis) and a new flowchart (Fig. 3 in the revised manuscript) to exemplify the applied processes.
- For example, the Authors use both Fire Weather Index (FWI) and the FFWI which is referred to as the Fosberg Fire Weather Index. Though details regarding the well-known FWi are provided, few are presented for the not-well-known FFWI. The descriptions of the indices should be similar. A more detailed description of the SPI along with the scale should also be provided. This subsection also lacks references. The same for the NDVI description. All methodology should be clearly stated in this section. It is very confusing to follow if the authors are using monthly data, daily data, which dataset, if they are using the fire season months (this should also be stated), and so on, for each section in the results. The MLE should be presented in this section and not in the results in my viewpoint..
The authors thank the reviewer for his/her suggestions/guidelines/comments and these directions are completely followed in the revised manuscript (see Section 2 in the revised and annotated manuscripts).
- All figures should also be revised. First, the legend presents two FWI there are 7 captions for 6 images, something is wrong (figure 2, for example). Also, the continuous scale makes the interpretation of the images very difficult since for example the color barrier for SPI between the negative and the positive values cannot be depicted. I strongly advise an interval scale for all figures.
The authors thank the reviewer for his/her suggestions and guidelines/comments. These comments have been completely addressed in the revised manuscript.
- *I find it very difficult to understand what the authors intend to achieve as solid results in section 3.2. . Due to the size of the study area and the differences in orography and vegetation, besides other aspects, performing a linear correlation will not give us scientifically robust and statistically significant results. Authors should perform correlation pixel-by-pixel and shows the patterns of the statistically significant pixels. This will show you the regions in which the correlations are significant besides their strengths.
The author’s comment has been addressed in the revised manuscript in Section 3.2 and the new Section 3.3: “Influence of meteorological variables on FWI on a daily /single point basis”.
- Typically, FWI is computed daily thus allowing civil protection authorities to implement the necessary measures to prevent major wildfires. Therefore, the study of the monthly variation of the FWI does bring some insight into its climatology. Let me recall that the meteorological conditions vary daily, thus impacting some of the components of the FWI, which in turn will impact the FWI itself. In section 4, the authors state: ‘The netCDF files containing in formation about the FWI index and its component sub-indices came in a monthly format for each corresponding time period and were merged to capture the entire period (March – October) in each dataset. Although time periods are not identical the results seem to be similar regarding average values of the index for the fire seasons of each time period, and they seem to follow a similar spatial pattern, concentrating their higher values in the southeast of the country.’ Seem to follow does not sound like a very scientific statement. Furthermore, the spatial distribution (pixel by pixel/coordinates) is also not coincident. Again, ‘Tendencies and patterns for all NASA datasets (GPM-Final, GPCP, TRMM long term mean) are similar.’ in lines 444/445. There are statistical techniques that ensure that the tendencies (or trends) and patterns are indeed statistically significant. This for me is a huge problem in the study presented in this manuscript. These statements are also presented in section 5 Lines 492/493 ‘When it comes to future projections of the index, we note increasing trends regarding the mean and max values of the FWI in both time periods for both RCPs.’ Authors cannot state this just by looking at the spatial patterns or by averaging FWI values for the study area and fire season. Means tend to be highly sensitive to outliers, but also to strongly smooth the values in a series. The trends should also be tested for their statistical significance.
The authors thank the reviewer for his/her comments and suggestions. These comments have been addressed in the revised manuscript. The Mann–Kendall trend test is used to assess statistical similarities between the employed indices. Two new paragraphs (Lines 715-758 in the revised manuscript) and Table 9 have been added in Section 5 to assess and discuss the application of the Mann–Kendall trend test at future periods. Furthermore, a completely new Section (Section 6: Discussion) has been added in the revised manuscript resolving all the above issues and comments made by the reviewer.
- A discussion should be provided, and the conclusions should also be readdressed since the ‘take-home’ message is not scientifically robust nor clear from my viewpoint.
The author’s comment has been addressed in the revised manuscript (Sections 6, 7 and 8 and in lines 806-927).
- Minor comments:
RCP: this acronym is not defined in the text. Also, when using it, it should be preceded by the word under, e.g. under RCP 4.5…
Line 335: it is not correct to say FWI is meteorologically derived… FWI is derived from meteorological variables or parameters…
Line 339/340: This is not scientifically correct. One can manipulate the color scales to match the patterns…
Line 422: [10] found… this does not follow the MDPI author’s guidelines. It should be written as: Karali et al. [10] found … The same comment should be applied throughout the text.
All references should be revised, there are incomplete references. Also, I advise authors to read other works regarding FWI, namely in Italy, Spain, and Portugal.
Captions: Captions should be revised, again under RCP 4.5 or 8.5. Figures 7 and 8: for me is not clear if the authors use daily values from January 1st, 2041 until December 31st, 2060, or daily values between the fire season. This should be clearly stated previously in the methodology section, but also in the captions.
All author’s minor comments have been addressed in the revised manuscript.
Finally, we are grateful to the reviewer for his/her constructive comments and suggestions that helped us to improve the manuscript in its revised form.

Reviewer 2 Report
Dear and respective authors,
The manuscript entitled Contribution to the study of forest fires in semi-arid regions with the use of Canadian Fire Weather Index. Application in Greece represent valuable study which explores in depth the spatiotemporal patterns of the FWI system in Greece, a south European Mediterranean region prone to forest fires due to extreme weather events influenced by climate change.
Although the presented work with valuable methodology and results deserve to be considered for publishing in the Climate scientific journal, it still has some issues needed to be addressed before this step. Below is the list with my suggestions for manuscript enhancement.
1. The title of the manuscript should be slightly modified into: Contribution to the study of forest fires in semi-arid regions with the use of Canadian Fire Weather Index. Applications in Greece.
2. In chapter 1. Introduction (page 1, line 26-113) respective authors should expand the part related to the literature review of international papers that used similar methodological approaches. In the current state this chapter is rather poorly presented, thus authors should expand this part significantly.
3. On page 3, subchapter 2.1 Study area (line 124-144): physical map of the investigated area is missing. Also, climate maps with air temperature, precipitation, wind speed should be included as a supplementary part of this figure. More information about climate properties are needed within this section.
4. An additional figure: "Flow chart with all the procedures and methods used in this research" should be added since it can illustrate a complex number of methodological steps and procedures used in this research. Respective authors can add this figure after line 316, on page 8 stating that: "All procedures and approaches used for the purpose of this research are presented in the flow chart given in Figure XXX".
5. Page 9, Figure 2. Please remove the blue background from the maps since it reduces its readability.
6. On figures 4, 5, 6, please remove the titles from the figures since they are clearly stated in the figure captions.
7. Discussion is completely omitted. Respective authors should enhance chapter "3. Results" and make it chapter "3. Results and discussion" by comparing the obtained results of this study with similar studies that used FWI and associated indices for the Mediterranean and region of South-East Europe.
8. Before chapter 6. Conclusions, authors should make one small additional chapter "Limitations and Future Research Directions" in order to outline advantages and disadvantages/limitations of the applied methodology.
9. Reference list should be enhanced, 29 references related to this topic are not sufficient to back up results obtained by the respective study.
10. It is highly advisable to respective authors to proofread the article after the corrections since there are typos within the main text.
After the above mentioned suggestions for improvement are implemented into the manuscript, I highly suggest to the Editorial Board of Climate scientific journal to consider it for publication. It certainly has great potential considering the research field and methodological approaches.
Kind regards!
Author Response
Comments from the Reviewers
Reviewer #2 - Comments
The manuscript entitled Contribution to the study of forest fires in semi-arid regions with the use of Canadian Fire Weather Index. Application in Greece represent valuable study which explores in depth the spatiotemporal patterns of the FWI system in Greece, a south European Mediterranean region prone to forest fires due to extreme weather events influenced by climate change. Although the presented work with valuable methodology and results deserve to be considered for publishing in the Climate scientific journal, it still has some issues needed to be addressed before this step. Below is the list with my suggestions for manuscript enhancement.
The authors thank the reviewer for his/her constructive and useful comments. All comments made by the reviewer have been addressed and the mistakes have been corrected in the revised paper (see revised manuscript and annotated manuscript). The reviewer comments are shown in “plain text” and the authors’ response in the text with italics.
- The title of the manuscript should be slightly modified into: Contribution to the study of forest fires in semi-arid regions with the use of Canadian Fire Weather Index. Applications in Greece.
The author comments have been partially addressed in the revised manuscript. We believe that the term “Application” (instead of Applications as the reviewer suggested) in the tittle maybe is more appropriate in our study.
- In chapter 1. Introduction (page 1, line 26-113) respective authors should expand the part related to the literature review of international papers that used similar methodological approaches. In the current state this chapter is rather poorly presented, thus authors should expand this part significantly.
We thank the reviewer for his/her constructive comment. Indeed the section 1: Introduction is completely rewritten according to reviewer guidelines (lines 42-148 in the revised manuscript. Furthermore, the cited references are now 69 instead of 29 (40 new references have been added and discussed in the manuscript). Finally, new scientific elements are incorporated in the manuscript and supported with statistical analysis and subsequent discussion and the revised manuscript has been completely redesigned and restructured, accordingly (see revised and annotated manuscript).
- On page 3, subchapter 2.1 Study area (line 124-144): physical map of the investigated area is missing. Also, climate maps with air temperature, precipitation, wind speed should be included as a supplementary part of this figure. More information about climate properties are needed within this section.
We thank the reviewer for his/her constructive comment. This comment is addressed in the revised manuscript (New Figure 1). Additional information like mean annual Precipitation, Temperature and Wind Speed) is not provided due to paper length limitations.
- An additional figure: "Flow chart with all the procedures and methods used in this research" should be added since it can illustrate a complex number of methodological steps and procedures used in this research. Respective authors can add this figure after line 316, on page 8 stating that: "All procedures and approaches used for the purpose of this research are presented in the flow chart given in Figure XXX".
The reviewer’s comment has been addressed in the revised manuscript. A new flowchart (Fig. 3) has been added and discussed in the revised manuscript according to reviewer recommendations.
- Page 9, Figure 2. Please remove the blue background from the maps since it reduces its readability.
The reviewer’s comment has been addressed in the revised manuscript. The Figure (Figure 3 in the revised manuscript) is redesigned according to reviewer guidelines.
- On figures 4, 5, 6, please remove the titles from the figures since they are clearly stated in the figure captions.
The reviewer’s comment has been addressed in the revised manuscript. The Figures are redesigned according to reviewer guidelines (see Figures in the revised manuscript).
- Discussion is completely omitted. Respective authors should enhance chapter "3. Results" and make it chapter "3. Results and discussion" by comparing the obtained results of this study with similar studies that used FWI and associated indices for the Mediterranean and region of South-East Europe..
The reviewer’s comments have been addressed in the revised manuscript. The respective sections have been completely restructured and rewritten according to reviewer guidelines.
- Before chapter 6. Conclusions, authors should make one small additional chapter "Limitations and Future Research Directions" in order to outline advantages and disadvantages/limitations of the applied methodology.
The reviewer’s comments have been addressed in the revised manuscript. A new Section has been added in the revised manuscript according to reviewer guidelines (see Section 7 Limitations and Future Direction and Lines 843-863 in the revised manuscript).
- Reference list should be enhanced, 29 references related to this topic are not sufficient to back up results obtained by the respective study.
The reviewer’s comments have been addressed in the revised manuscript. The cited references are now 69 instead of 29 (40 new references have been added and discussed in the manuscript).
- It is highly advisable to respective authors to proofread the article after the corrections since there are typos within the main text.
The reviewer’s comments have been addressed in the revised manuscript. All typos and mistakes have been corrected according to reviewer recommendations.
.
- After the above mentioned suggestions for improvement are implemented into the manuscript, I highly suggest to the Editorial Board of Climate scientific journal to consider it for publication. It certainly has great potential considering the research field and methodological approaches.
We thank the reviewer for his/her positive attitude in our study and his/her constructive comments and suggestions that helped us to improve the manuscript in its present form.

Round 2
Reviewer 2 Report
Dear and respective authors,
After thorough inspection of the manuscript I can see that it was enhanced significantly. I only advise the authors to check the main text for technical errors during the proofread phase. In general I advise the Editorial Board of Climate scientific journal to accept it for publication.
Congratulations and kind regards!